

# Do users benefit from additional information in support of operational drought management decisions in the Ebro basin?

Clara Linés[1], Ana Iglesias[2], Luis Garrote[3], Vicente Sotés[2], and Micha Werner[1,4]

[1]IHE-Delft, Department of Water Science and Engineering, Delft, the Netherlands
[2]Technical University of Madrid, Department of Agricultural Economics and Social Sciences, Madrid, Spain
[3]Technical University of Madrid, Department of Civil Engineering: Hydraulics, Energy and Environment, Madrid, Spain
[4]Technical University of Madrid, Department of Plant Production, Madrid, Spain

**Correspondence:** Clara Linés (c.lines@un-ihe.org)

**Abstract.** We follow a user based approach to examine operational drought management decisions and how the role of information such as from remote sensing can be assessed. First we consulted decision makers at basin, irrigation district and farmer scale in the Ebro basin to investigate the drought related decisions they take and the information they use to support their decisions. This allowed us to identify the courses of action available to the farmers and water managers, and to analyse their

choices as a function of the information they have available to them. Based on the findings of the consultation, a decision model representing the interrelated decisions of the irrigation association and the farmers was built with the aim to quantify the effect of additional information on the decisions. The modelled decisions, which consider the allocation of water, are determined by the expected availability of water during the irrigation season. This is currently primarily informed by observed reservoir level data. The decision model was then extended to include additional information on snow cover from remote sensing. The addi-

tional information was found to contribute to better decisions in the simulation and ultimately higher benefits for the farmers. However, the ratio between the cost of planting and the market value of the crop proved to be a critical aspect in determining the best course of action to be taken and the value of the (additional) information. Risk averse farmers were found to benefit least from the additional information, while less risk averse farmers can benefit most as the additional information helps them take better informed decisions when weighing their options.

# 1   Introduction

Water managers and farmers regularly take decisions to make the most of the available water resources. Information on the availability and variability of the resource is essential to allow these decision makers to choose among the actions available to them, especially as water becomes scarce, for example during drought events. Improved information on the availability of water can then potentially lead to a more effective management and can therefore contribute to the mitigation of the impacts of

drought events.

    In situ meteorological and hydrological measurement networks have long served to inform these decisions, often providing accurate water resources observations at high temporal resolution. In addition, the potential of Earth Observation (EO) from satellites to support water management has also been widely recognised (Famiglietti et al., 2015; Fernandez-Prieto et al., 2012).





The availability and quality of EO datasets has been continuously improving during the last decades, providing an increasingly relevant source of globally consistent data that can be used to complement in situ data.

However, the increased quality and availability of information does not necessarily translate directly into benefits due to better decisions. How the information is used and distributed also plays a critical role (Williamson et al., 2002). It is the

capacity of the user of information to change the course of action as a result of new information being available to them that largely impacts the value of that new information (Macauley, 2006).

A good understanding of the role that information plays or could play in supporting decisions, as well as the benefits derived from using that information to inform the decision, is useful both for the users and the data providers, and helps improve the connection between these two groups. Onoda and Young (2017), in their conclusions established on a series of analyses

on the contribution of EO datasets in addressing environmental problems from a policy point of view, recommend more stakeholder oriented studies of the value of these data and the quantification of the benefits of this data through comparisons with current tools. The assessment of the role and impacts of remote sensing products is expected to help in fully achieving the potential of the products, maximising the socioeconomic and environmental benefits and contributing to justify the investment in developing and improving the products.

An example of a stakeholder oriented approach to assess the value of satellite based information in support of water management is presented by Bouma et al. (2009). They develop a framework to measure the benefits of satellite based observations. The framework, which is based on Bayesian decision theory and expert consultation, is applied to water quality management in the North Sea (Bouma et al., 2009) and to coral reef protection (Bouma et al., 2011).

Macauley (2006) reviews studies on the value of information in Earth science applications, classifying the techniques that

are used or that are potentially useful in three groups: studies that measure the value by gains in output or productivity; studies based on hedonic pricing, in which the value is inferred from models based on wages and housing prices; and studies that consider the willingness-to-pay. The main example of the first group of techniques are the studies that relate farm profits and weather information, especially in relation to weather forecasts. Early studies explore simplified cases of decisions such as whether to plant or to leave cultivable land fallow (Brown et al., 1986), or on what crop to plant (Wilks and Murphy, 1986).

From the user's perspectives, the optimal choice for a decision to be made is to take the course of action that results in the highest expected utility, which is defined as the weighted sum of the outcomes of the possible actions and the probability of a given state of nature such as a reduction in the available water resource. Clearly this includes undesirable outcomes, where an action is taken based on an expected state of nature that does not materialise. Additional information is then considered to have value if it can improve the advance knowledge on the probabilities of the different possible states of nature occurring, thus

allowing the user to take a better informed decision. A commonly used approach to evaluate the value of advance information, such as information provided through advanced warning, is the cost-loss framework (Zhu et al., 2002; Mylne, 2002; Roulin, 2006; Verkade and Werner, 2011). In the case of warnings for adverse events this is the ratio of the costs of taking protective action to the losses incurred if the action is not taken. This framework has also been extended to water resources management decisions, such as in (Quiroga et al., 2011), who analyse the value of climate projections for the water manager decision to apply

measures to reduce the water demand in the Ebro basin. The cost-loss framework does assume a strictly rational behaviour of





users in weighing the costs and probability of losses, but has its limitations as different users may make different decisions depending on their levels of risk averseness, though this can be incorporated through a function of risk aversion (Quiroga et al., 2011; Matte et al., 2017).

In this paper we follow a user based approach to examine the operational decisions that stakeholders such as farmers and reservoir operators take within the context of water resources allocation during droughts. The proposed framework first identifies the decisions stakeholders take, and the information they use to inform those decisions, through interviews. A decision model is then established and applied to emulate the decision process using the currently available information, as well as how additional information contributes to improving the decisions made. This study is part of a broader research aimed at assessing the usefulness of global datasets to support decisions at the basin or sub-basin scales.

## 2  Methods

### 2.1  Study area

We explore the role of information in drought related decisions in the Ebro basin. The Ebro Basin is the largest in Spain (85,600 km$^2$), and is a highly regulated basin with 125 reservoirs (>1 Mm$^3$) and a total storage capacity of approximately 8,000 Mm$^3$. These reservoirs are used primarily to supply water to more than 900,000 ha of irrigated agriculture and 360 hydro-electrical plants (CHE, 2015).

The larger irrigation districts are located in the north-east of the basin (Figure 1). We have selected one of these, the irrigation district supplied by the Aragón and Cataluña channel (Canal de Aragón y Cataluña, CAyC) to examine the decisions made at the sub-basin scale.

Over 90% of the water provided by CAyC is used for irrigation. The water it supplies is sourced from three reservoirs (Barasona, San Salvador and Santa Ana), and it is supplied to an irrigated area of around 98,000 ha. Two zones can be distinguished in the irrigated area: an upstream zone that can only be supplied from the Barasona reservoir as well as from the recently inaugurated San Salvador reservoir, and a downstream zone that can be supplied from all three reservoirs. These zones are 54,000 and 44,000 ha in size, respectively. The main crops grown in the area are fruit orchard (apple, pear, peach and nectarine) and extensive herbaceous crops, mainly maize, alfalfa and barley. The area cropped with wine vine surface is increasing, though it is still somewhat localised (CHE, n.d.).

### 2.2  Approach

#### 2.2.1  Investigating stakeholders' decisions

As the utility of information strongly depends on the particular details of the decisions and how information is used to support these, we first consulted decision makers at the basin (Confederación Hidrográfica del Ebro, CHE), irrigation district (Comunidad de Regantes del Canal de Aragón y Cataluña), and farmer scale to better understand their decision processes, their





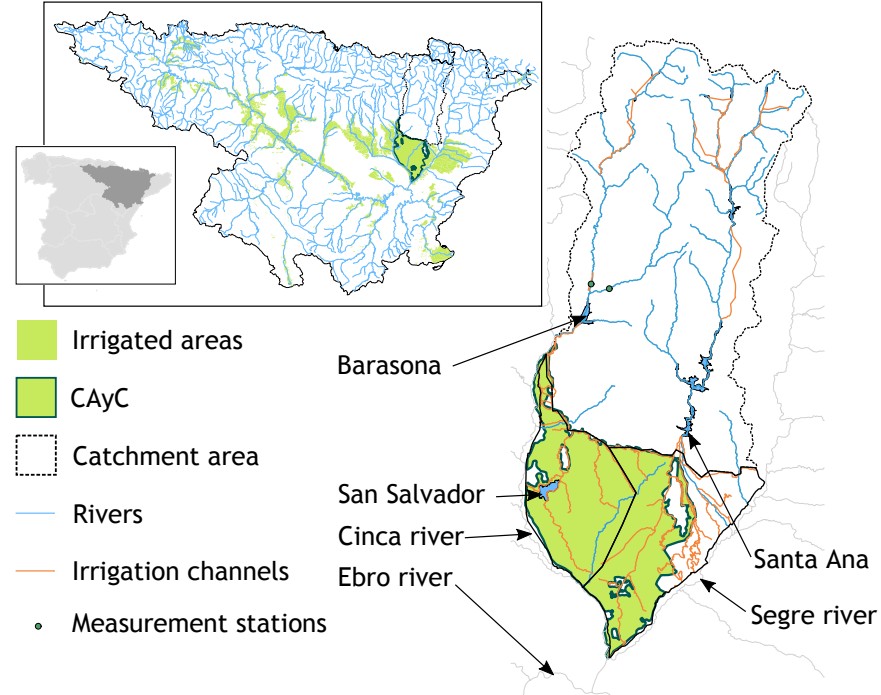

**Figure 1.** Canal de Aragón y Cataluña: irrigated area and catchment.

information needs, and how they use information to support the decisions they make. For this analysis we focused on decisions regarding the allocation of water resources, in particular during drought, when curtailments may be applied (Figure 2).

Several studies have highlighted the significance of taking stakeholders into consideration when implementing a strategy (e.g. Bryson, 2004; Eden and Ackermann, 1998; Freeman, 1984) and this concept has been applied extensively to water resources

management (see for example Iglesias and Garrote, 2015). Here we consider as stakeholders the different groups that have the capacity to modify the amount of water to be used for irrigation either by deciding the volume of water to be supplied, or by deciding the area and type of crops planted; thus effectively determining the irrigation demand. Considering this rationale, we included two types of stakeholders: reservoir operators and farmers. We make the assumption that these two groups are uniform within themselves, and that public opinion or interest groups such as environmentalists do not alter the decision. This

is clearly a simplification, which could be overcome by carrying a large survey. We consider such a survey as being beyond the scope of this study.

The selection of the participants for the interview is a critical strategic decision (Iglesias and Garrote, 2015). The choice of individuals is guided by the potential unique information that they can provide to the study in relation to water management decisions under drought conditions. The group of farmers has an unequivocal and unanimous interest in avoiding water short-

ages and ensure irrigation production. All farmers are included in an Irrigation Association that provides them with services of



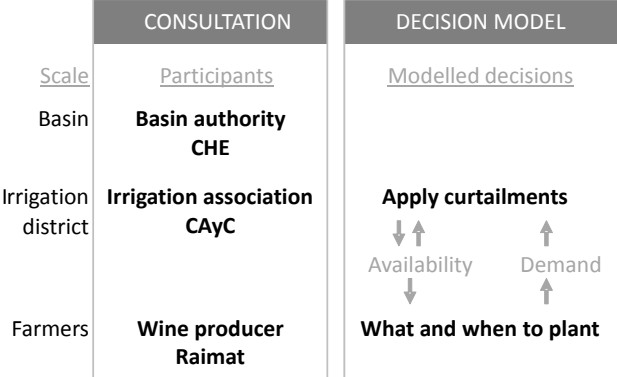

**Figure 2.** Research phases and spatial scales.

high technical skill and tactical advice on the use of water. The technical advisors at Irrigation Association were selected for an interview as representatives of this position and the collective interests. The basin authority (CHE), responsible for defining measures to be implemented during drought and water shortages, was also interviewed.

Details on water management and possible adjustments of decisions that the stakeholders may take in view of water shortages were collected by means of semi-structured interviews. This is a clear methodological choice to collect additional knowledge not directly articulated in the interview guideline (Harrell and Bradley, 2009). In our case, we wanted to obtain information related to the management choices in drought situations and the barriers and incentives to implement them that are only known in practical terms. The semi-structured interview also gives the possibility to incorporate additional discussion topics, not envisioned by the research team in the guideline.

## 2.2.2 Modelling the decision

The findings of the consultation phase were then used to build a model of the decisions at irrigation district scale, in order to test and quantify the effect that additional information has on the analysed operational drought management decisions. The decision model was built in R (R Core Team, 2016).

In the year to year planning of water allocation and crops to plant, the basin authority is of lesser relevance in the decision model as they are responsible for the longer term planning through the basin hydrological plan, and for the development of the drought management plan, which though it provides guidelines on measures to be taken during drought, does not go to the detail of the operational decisions to be taken on water allocation. These are taken by the reservoir operators (part of the irrigation association, CAyC), while decisions on what and how much to plant are taken by the farmers. The model represents these two decisions, as well as the interaction between them.

Expected availability of water during the irrigation season is the main variable used by both the reservoir operators and the farmers to inform their decisions. Information on the availability of water can, however, be obtained from different sources.





Currently the main source of information that is used is the volume stored in the reservoirs, obtained through observations of the reservoir levels. Both reservoir operators and farmers indicated that they may also consider the available water resource in the snowpack in the headwaters upstream of the reservoirs, though there are currently no systematic observations of this resource that are formally included in the decision process by either. Satellite images can, however, routinely provide estimates of this

resource. Two situations with different input information were therefore simulated: the expected water resource availability as informed by the reservoir levels alone, and the expected availability based on the reservoir levels with the addition of satellite based data on snow cover in the headwaters. In both cases the outputs of the decision model are the areas of different crops that are planted during the irrigation season and the water demand that this cropping schedule results in; the curtailments applied; and the resulting benefit for the farmers in each season. Crop water demand and yield were modelled with AquaCrop-OS

(Foster et al., 2017) and CROPWAT 8.0 (FAO, 2000).

### 2.2.3  Quantifying the effect of additional information

Output benefit values using either of the two information scenarios were evaluated against the value (Val) of uninformed decisions and decisions under perfect information following the usual form of skill scores (Stanski et al., 1989):

$$RV = \frac{Val_{information} - Val_{uninformed\ decision}}{Val_{perfect\ information} - Val_{uninformed\ decision}} \tag{1}$$

The relative value ($RV$) is therefore a score between $-\infty$ and 1, with $RV = 1$ meaning that the information is perfect and $RV \leq 0$ meaning that the information does not contribute to improving the decisions made.

The value of perfect information was calculated by selecting for each season the best performing course of action given the observed water resource availability, while the value of the uninformed decisions was defined as the result of selecting for all years the course of action that performs best based on the average water resource availability for the whole period.

## 2.3  Model input data

**Reservoir and meteorological data** In situ data on reservoir levels was obtained from the automatic measurement stations (SAIH, Automatic Hydrologic Information System). These data are available from http://sig.mapama.es/redes-seguimiento/.

Reservoir volume data for Barasona reservoir and river flow data from the stations at the upstream tributaries (stations located at Graus on the Ésera river, and at Capella on Isábena river, Figure 1) was used to estimate the availability of water during the

season. We focus on the Barasona reservoir as it is the levels in this reservoir that triggers the restrictions in the area supplied by CAyC. SAIH provides data for Barasona reservoir from 1931 to September 2014, though there are some data gaps in the first decades. The reservoir was enlarged in 1972 to a capacity of 84.71 hm$^3$ and we therefore consider only the values after that year.

In addition, daily precipitation and temperature data, and monthly relative humidity data from the meteorological station

located just outside the basin at the University of Lleida (station 9771C) was used to provide meteorological inputs to the crop model. Data from this station is available from 1983 through 2014 and can be obtained from https://opendata.aemet.es/centrodedescargas/productosAEMET



**Snow cover data** MODIS 8-day snow cover 500 m grid data (MOD10A2; Hall et al., 2006) was used to calculate the percentage of snow cover in the headwaters of the reservoirs (Figure 1) as an additional source of water availability information. This dataset covers the period from 26 Feb 2000 to the end of 2016 and was downloaded from the EartH2Observe Water Cycle Integrator (wci.earth2observe.eu).

# 3   Results

## 3.1   Investigating stakeholders' decisions

### 3.1.1   Confederación hidrográfica del Ebro (CHE)

The main operational decisions that the Ebro river basin authority (CHE) takes regarding drought are the declaration of drought conditions and the allocation of water in emergency situations. To guide these decisions, CHE defined a drought manage-
ment plan in 2007 (CHE, 2007), which was the first of its kind in Europe. It is a very comprehensive plan and links hydro-meteorological indicators to drought severity levels. The decision to declare drought is informed by a set of indicators defined in the plan. These indicators, built from measurements from the dense network of in situ automatic stations, are used to detect hydrological drought conditions with different levels of severity. The plan establishes the main indicator to be used for each of the areas of the basin. Where possible water stored in the reservoir is used as main indicator, since it is considered the most
robust option. In other cases 3-month water flow or groundwater levels are used as indicator.

In their opinion, the declaration of drought is currently well informed and therefore they consider that additional information would be more useful after the declaration, when conditions must be monitored closely and decisions such as selecting the most cost-effective alternative sources of water or how to secure sufficient water to guarantee environmental flows must be taken. For these decisions timing is critical and they point out that information should be available with a maximum delay of
one week to be useful for decisions. In addition, they showed particular interest in remote sensing derived snow data to support the quantification of water availability in the basin.

### 3.1.2   Canal de Aragón y Cataluña (CAyC)

General Irrigator Associations such as Canal de Aragón y Cataluña (CAyC) are responsible for the distribution of water from the reservoir to the users. In drought situations they can decide to introduce restrictions to irrigation water quotas. The decisions
they take on the application of these restrictions are informed by the availability of water in the reservoirs that feed the irrigation canal system.

The main decisions that CAyC take in relation to drought is to apply restrictions (curtailments) to the maximum amount of water that irrigators can request. They take this decision when they consider that the available water resource is insufficient to reach the end of the irrigation season if full irrigation supply to meet demand is maintained. They can also decide to move
water among the three reservoirs in the area. When restrictions are necessary, these are applied to all users independently of



the reservoirs that they can be supplied from to ensure curtailments are applied equitably across the district. However, when water is scarce, priority is given to the fruit orchards and vines to ensure their survival.

To take their decisions, the reservoir operators need information both on water availability and the expected demand until the end of the season. In the interviews, they indicated that they consider that they are well informed on the availability of water given the levels in the reservoir. However, the information on water demand is limited. The difficulty of knowing the demand is due to the fact that they lack information on what crops farmers are planning on cultivating that year, and especially if the farmers will decide to plant a second crop, thus increasing the demand towards the end of the season. Currently they use historic data to estimate the demand. They are also conducting studies on the feasibility of obtaining this information from remotely sensed NDVI data (described in Casterad Seral, 2015; Quintilla et al., 2014), and although this will provide useful information on the current crops, it will not provide information on the future plans of the farmers.

Unlike the managers at the basin scale, CAyC indicated that they consider additional data on the snow cover in the headwaters to be of little use in quantifying the available resource. They argue that they tend to be cautious when accounting for snow in the estimation of total availability, since the reservoir capacity is rather small and therefore the possibility to store snowmelt runoff depends very much on the melt rate.

### 3.1.3  Farmers in the Canal de Aragón y Cataluña irrigated area

During the consultation, CAyC also provided details on the types of farmers present in their supply area as well as on the decisions these farmers take on what to plant. Typically, the proportion of crops planted is fruit orchards and vines for roughly a third of the area, alfalfa for another third, and variable crops for the remaining third. These variable crops are mostly winter cereals and maize. The cropping schedule adopted by farmers is mostly either a single crop of long cycle maize or a winter cereal; or a double crop in which a winter cereal is followed by short cycle maize. The selection of one or the other by the farmers depends on the expected water availability, and is currently mainly informed by the water level in the reservoir. CAyC shares this information with them in the form of biweekly reports. Conversely, the decisions farmers take in terms of what crop to plant, and if they plan to plant a double crop, determines the demand for the season, and will therefore also have an impact on the decision to apply curtailments that are taken by the CAyC.

A prominent farmer in the supply area of CAyC is the Raimat wine producer, who also participated in the consultation process. They provided details of their information use for water resources management. Their parcels extend over 3,200 ha and are highly technified, with extensive use of detailed information. In addition to in situ measurements and meteorological station data, they use Landsat satellite data and perform flight campaigns to acquire spatial NDVI and thermal data. The thermal data is used to estimate the leaf water potential and, together with temperature data, calculate a crop water stress index. This information on crop condition is used to detect spatial differences in the crops and make the most of the limited water, select the optimal moment for irrigation and prevent plagues; as well as ensuring the production is as uniform and controlled as possible.

The decisions they take are already based on high resolution data, and therefore additional medium resolution global data are not likely to be a valuable contribution to this type of user. However, this extensive use of information is not representative of all the farmers in the basin and other farmers may indeed benefit from additional information.




## 3.2 Modelling the decision

Two of the drought related operational decisions described by the stakeholders were selected to be modelled. These are the decisions of the farmers on what to plant and the decision of the irrigation association to apply curtailments to the amount of water that farmers can request when this is considered necessary to avoid finishing the supply before the end of the irrigation

season. The model represents both decisions as well as the interaction between the two.

### 3.2.1 Farmer Decision: Crop areas

The farmers have a number of possible crop alternatives for each irrigation season. In this part of the model we simulate the decision of the farmers to follow one of the possible courses of action available to them. The result of the decision is the planted area of the selected crops. Since fruit orchards, vines and alfalfa crops last for several years, their approximate areal extent is

known and is considered as constant in the model. The farmer decision model therefore focuses on determining the variable areal extent of maize and winter cereal, as well as the decision whether to plant a single crop or also a second crop. In the model, barley is selected to represent the winter cereal crop since it is the most common winter cereal crop in the area.

The courses of action represented in the model consist of a series of decisions made during the irrigation season. The possible actions are depicted in Figure 3, which shows the choices that can be made at each decision stage in the calendar.

At each decision point, the option that farmers would prefer to take is indicated by a blue letter, while the non-preferred decision(s) is marked with a red letter. The choices and the calendar are based on the information provided by stakeholders we interviewed, supported by literature sources (Espluga Trenc, 2016; Gil Martínez, 2013; Lloveras et al., 2014). The model considers two types of farmers with different options available to them: technified farmers managing large plots and smaller scale farmers. In the model, technified farmers (marked with T1 in Figure 3) can support a double crop, and this is always their

preferred option because of its higher productivity. However, in years of low water availability (red A in the figure) they may decide to leave the land fallow instead of planting a second crop. Smaller scale farmers (T2) can only manage a single crop and in this case long cycle maize is the most productive option. In years of low water availability, their decision will depend on the level of risk they are willing to take. The safest option to secure a crop is to plant long cycle barley at the beginning of the season, but they can also decide to wait for conditions to improve; taking the risk of having to leave the land fallow if there

is no improvement. If water availability increases by February, they can decide to plant long cycle maize. If availability is still low, they can secure a crop by planting a short cycle barley (a less productive option than the long-cycle version) or they can decide to wait longer. In April, they can still plant a long-cycle maize crop if availability has improved, but, if it has not, then it is too late to plant barley and they have no other option than to leave the land fallow.

The choice of a particular course of action in the model is based on the water availability at the moment of the decision, and

is used as an indicator of the expected availability of water during the remaining season. For the decisions taken before the start of the irrigation season (November and February), the availability is based on the observed volume in the reservoir since these decisions have no influence on the level of the reservoir until the actual start of the irrigation season. For the decisions after the start of the irrigation season (April and May) a simulated volume is used. The simulated volume is based on the observed



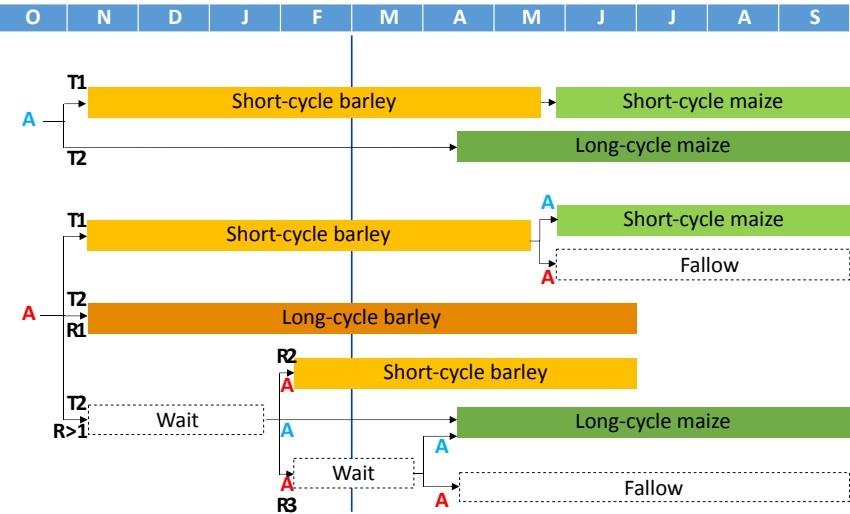

**Figure 3.** Crop options considered in the model for farmers. Blue and red As represent respectively good and poor water availability at the moment of the decision. R1, R2 and R3 mark the different paths that can be followed depending on the risk the farmers are willing to take, with R1 being the most risk averse and R3 the least risk averse. The blue vertical line marks the start of the irrigation season.

volume at the beginning of the irrigation season and the accumulated inflow from the beginning of the irrigation season until the decision date (Table 1). From the start of the irrigation season the decisions taken will have an influence on the level of the reservoir and therefore observed levels are not representative as an input for further decisions. Note that the barley crop is not irrigated, and therefore does not pose a demand on the available water resource. However, as can be seen in Figure 3 the
choice for planting a barley crop will have implications for the available options to be made later in the season, and therefore indirectly influences the irrigation demand.

### 3.2.2  Reservoir operation decision: Water restrictions

Every two weeks during the irrigation season (March-October), the decision on whether to apply curtailments to the maximum amount of water that irrigators can apply for is re-examined. This decision requires an estimate of the total amount of available
water during the season and the total water demand. To estimate the total amount of water that will be available during the season, the inflow into the reservoir from the beginning of the hydrological year in October up to the week of the decision is compared with the percentiles of historic data (Table 1). Data of accumulated inflow in the reservoir is preferred to reservoir levels as input information for the reservoir operator decision model to avoid the influence of the actual decisions of the managers and simulate them independently. The percentile curve in which the value of the current year is positioned is used to
sample from the climatological record the projection of the inflow series into the reservoir until the end of the season.





**Table 1.** Model decisions and parameters that inform them. A description of the abbreviations is included below. The period considered for each parameter is given in between square brackets. The colour indicates the availability and source of the information.

| Decision | Informed by | Requires | Output |
|---|---|---|---|
| Farmer decision (FD) [before s] | $V$ | Reservoir level data | Crops planted [before s] |
| Farmer decision (FD) [after s] | $V$ [i] | $F$ [h:i] , $D$ [s:i] | Crops planted [after s] |
| | | $D$ [s:i] = $CS$ x $CD$ | |
| | | $CS$ ← previous FDs | |
| | OD [s:i] (curtailments) | See OD below | |
| Reservoir operator decision (OD) [s:n] | $F$ [h:n] | $F$ [h:i] , $P_F$ [i] | Curtailments |
| | $D$ [s:n] | $D$ [s:i] | |
| | | $D$ [s:i] = $CS$ x $CD$ | |
| | | $CS$ ← $FD$ | |
| | | $D$ [i:n] ← crop model (avg. 2000-2014) | |

**Abbreviations and colours**

| Parameters | Period [] | Availability of the information |
|---|---|---|
| F – Inflow | s - start of irrigation season | known to the decision maker (from model) |
| D - Demand | i - current decision date | known to the decision maker (from data) |
| V - Volume | n - end of season | unknown to the decision maker (estimated) |
| CS - crop surfaces | h – start of hydrological year | |
| CD - crop demand | | |
| P – percentile | s:i – from s to i | |

The total water demand for irrigation until the end of the season is calculated as the sum of the demand until the decision day and the expected demand from the decision day until the end of the season (Table 1). The first is the product of the crop surfaces already planted, which is the output of the farmer decision model, and the resulting crop demand obtained from the crop models using observed meteorological data. The latter is unknown to the managers. In the model an average demand per
5    unit area of crop calculated with the crop model data for the period 2000-2014 is used as an estimate to inform their decision. This is a simplification, as the demand to the end of season will depend on the expected climatological conditions and the crop surfaces planted. The actual demand up to the decision day could be used to provide an indication of the expected demand until the end of the season. However, while that could provide an indication of the climatological conditions, the decisions taken by farmers on crops to plant at future decision moments are unknown.



When the estimated total amount of available water during the season is insufficient to fulfil the total demand, curtailments are applied. Conversely, if in a later week the expected total available water is found to be enough to fulfil the total demand, restrictions are lifted.

The output of the farmer model is the areas of each of the crops planted each year by the farmers. These crop areas determine the demand in the reservoir operator decision model. The output of the operator decision model on the other hand is the curtailments posed on the water supplied to the farmers to the end of the season, which is determined every two weeks. These curtailments may reduce the yield of the crops if these are already planted and the demand cannot be satisfied, but will also influence the farmer decisions within the irrigation season. If curtailments are in force when the farmers are deciding what to plant (April and May), then the model assumes that the farmers consider water availability as not being good, leading to decisions commensurate with low water availability being taken. These decisions will consequently influence the demand.

### 3.2.3 Crop water demand and benefit

AquaCrop-OS was used to simulate barley and maize yields. These crops are the main focus of the analysis and they require a more detailed and flexible simulation to differentiate the different growing cycles and planting dates. Default parameters for maize and barley were adapted for the diverse cycles using data from Lloveras et al. (2014), Gil Martínez (2013) and Gutiérrez López (2011).

CropWat was used for alfalfa and fruit orchards, the crops that are considered to have a constant crop surface in the analysis. Default parameters were used but were adapted to the cropping calendar in the Ebro basin. Peach tree was selected as the representative fruit orchard crop. An irrigation calendar of 14 days was selected to match the reservoir operators' decision.

The percentage of reduction of crop yield was calculated as the maximum percentage of unsatisfied demand during the season. The reason for this is that when there is insufficient water farmers prefer to stop watering a part of the area, rather than applying insufficient water to the whole area. These percentages were calculated using the same biweekly time step of the operator decision. The areas in which irrigation was stopped were considered to have no yield and their contribution was subtracted from the full supply yield values derived from the crop models to obtain the final yield for each crop and year. Priority is given to the multi-annual crops, with the curtailments then being applied to the maize crops.

### 3.2.4 Model Options

**Availability Thresholds**

Thresholds are needed to define at what reservoir level, or combination of reservoir level and snow cover, the water availability is regarded by the farmers as good at the decision points. If the availability is above the threshold, then the farmer would follow the decision path associated to good expected availability of water (Blue letters in Figure 3), while if it is below then the alternative, poor expected availability path will be followed. These thresholds are currently not formally defined, and may also differ between farmers as individual farmers will assess water availability differently, depending on how risk averse they are.

Here we first test the model with a set of thresholds optimised to maximise the sensitivity (rate of true positives) and specificity (rate of true negatives). These are measures of the goodness of a binary classification that in this case refers respectively





to the points correctly classified as good or poor availability. To assess the performance of the classification the decisions taken with perfect information are used as a reference.

We explore the effect of different thresholds by testing 10 additional sets of thresholds, ranging from low availability to almost full capacity (35 to 80 hm$^3$), keeping the same thresholds at each of the four points where the farmers make decisions

during the season.

The effect of the additional information on the expected available water resource that is provided by the data on the snow cover is incorporated in the decision model by considering the expected contribution of snowmelt to the available water resource. When the snow cover is below a certain threshold, indicating lower than normal expected runoff from snowmelt, the farmers would require the reservoir level threshold to be higher to regard water availability as being good and thus follow the

higher water demanding path. The snow cover thresholds used for this test are determined in an optimisation step, where the reservoir level threshold is maintained at a high level, and the snow cover threshold is again determined using a goodness of fit measure of the binary classification of the decision points correctly identified as having good or poor availability. For the decisions taken in May the snow information was not considered since snow cover is already very limited in that period.

**Allocation Factor**

An allocation factor is applied to the accumulated inflow in the reservoir to obtain the proportion of the accumulated water that effectively reaches the crops. This factor accounts for water supplied to other uses, water losses due to evaporation, efficiency of the distribution network, and releases from the reservoir to the downstream river. The allocation factor determines the amount of water that is available for irrigation and therefore has significant influence on the decisions taken by the farmers and operators. As the true allocation factor is not known for the area, the sensitivity to this factor is tested by running the model

with different allocation factors.

To calculate the crop demand an irrigation efficiency of 80% is considered.

**Farmer Types**

The distribution of the types of farmer was kept constant for all the years and runs. The proportions of technified and smaller scale farmers was established as the mid-range of the yearly ratio between farmers sowing transgenic maize (considered as

technified) and farmers sowing conventional maize (considered as small scale farmers) observed in the area for the period 2010-2015 (Espluga Trenc, 2016; Gutiérrez López, 2016). This resulted in 65% of the area being exploited by farmers considered as technified, and 35% by small scale farmers. The proportion of risk aversion used in the model is R1=0.4, R2=0.3, R3=0.3, with R1 being the most risk averse and R3 the most risk acceptant.

Different distributions of farmer types would result in different levels of demand and therefore different optimal paths. The

proportion between technified farmers and smaller scale farmers also gives more weight to different decision moments. For example, the decision in May on whether to plant a second crop is only relevant for the technified farmers.

**Costs and benefits**

Planting costs and selling prices were used to calculate the value of the yield for the variable crops. The planting costs considered are 496 euro/ha for barley and 1807 euro/ha for maize (MAGRAMA, 2015) and the selling prices are 159 euro/1000

kg for barley and 171.3 euro/1000 kg for maize (Aragon Statistics Institute, n.d.). Average yields are 2.349,75 kg/h for rainfed



| path | Nov | Feb | Apr | Mar |
|------|-----|-----|-----|-----|
| 1 | 🟥 | 🟥 | 🟥 | 🟥 |
| 2 | 🟥 | 🟥 | 🟦 | 🟥 |
| 3 | 🟥 | 🟦 | 🟨 | 🟥 |
| 4 | 🟥 | 🟥 | 🟥 | 🟦 |
| 5 | 🟥 | 🟥 | 🟦 | 🟦 |
| 6 | 🟥 | 🟦 | 🟨 | 🟦 |
| 7 | 🟦 | 🟨 | 🟨 | 🟨 |

**Table 2.** Possible courses of action for farmers. The colours indicate the course that is followed: red – poor availability, blue – good availability, yellow – indifferent.

barley and 12.179,34 kg/ha for irrigated maize (MAGRAMA, 2015). No differences in price or cost between the varieties of a same crop type were considered, although the higher productivity of long cycle varieties results in these varieties being more profitable in the model.

### 3.3 Quantifying the effect of additional information

To quantify the effect of the additional information, the model was run for the two scenarios with different input information and for two reference scenarios: perfect information and uninformed decisions.

The value of perfect information and uninformed decisions was calculated by running the model for all possible courses of action represented in Figure 3. The seven possible options are summarised in Table 2, where the columns represent the four decision points and the colours the course that is followed. Blue and red indicate, respectively, that the good or poor

water availability option is followed. The points at which no decision is required are marked in yellow. This happens when previous decisions already determine the course of action for later months. Option 7 corresponds to the situation in which the availability of water is good at the beginning of the hydrological year so farmers select to plant the most productive crops already in November and no further decisions are required in the following months. The other six options correspond to situations in which the availability of water is not considered to be good at the beginning of the hydrological year. In options

3 and 6, the situation improves by February, so small scale farmers decide to plant the preferred option (long cycle maize) at this point and do not require further decisions. The difference between these two options results from the decision of technified farmers whether or not to plant a second crop in May. They will do this if they consider the availability of water to be good (option 6), otherwise they will leave the land fallow (option 3).

### 3.3.1 Farmer decisions and curtailments using perfect information

The selection of choices that can be made by a farmer that result in the highest benefits given perfect knowledge on the expected availability of water were identified for different water availability scenarios as a function of the proportion of the allocation factor. This factor determines the fraction of the total available water that reaches the irrigation area. The results are included in



**Table 3.** Option selected in the model by the farmers for each of the years of the period 2001-2014 (represented in the columns) in function of the available water determined by the allocation factor (AF). The numbers of the options refer to the alternatives included in Table 2.

| | Year | | | | | | | | | | | | | |
| AF | 01 | 02 | 03 | 04 | 05 | 06 | 07 | 08 | 09 | 10 | 11 | 12 | 13 | 14 |
|---|---|---|---|---|---|---|---|---|---|---|---|---|---|---|
| **1** | 7 | 7 | 6 | 6 | 7 | 6 | 7 | 7 | 6 | 7 | 6 | 6 | 7 | 6 |
| **0.8** | 7 | 7 | 6 | 6 | 3 | 6 | 7 | 7 | 6 | 7 | 6 | 6 | 7 | 6 |
| **0.6** | 7 | 6 | 6 | 6 | 1 | 4 | 7 | 7 | 6 | 7 | 6 | 6 | 7 | 6 |
| **0.55** | 7 | 4 | 6 | 6 | 1 | 1 | 7 | 7 | 6 | 7 | 6 | 5 | 7 | 6 |
| **0.5** | 7 | 3 | 6 | 6 | 1 | 1 | 7 | 5 | 6 | 6 | 6 | 3 | 7 | 6 |
| **0.475** | 7 | 2 | 6 | 6 | 1 | 1 | 6 | 4 | 6 | 6 | 4 | 2 | 7 | 6 |
| **0.45** | 7 | 1 | 6 | 6 | 1 | 1 | 4 | 4 | 6 | 4 | 1 | 1 | 7 | 4 |
| **0.425** | 7 | 1 | 6 | 6 | 1 | 1 | 3 | 4 | 6 | 4 | 1 | 1 | 7 | 4 |
| **0.40** | 7 | 1 | 6 | 6 | 1 | 1 | 2 | 4 | 1 | 4 | 1 | 1 | 7 | 4 |
| **0.20** | 4 | 1 | 1 | 1 | 1 | 1 | 1 | 1 | 1 | 1 | 1 | 1 | 1 | 4 |

Table 3. The first row (AF=1) represents the hypothetical situation in which all the water that enters the reservoir is available to the farmers to irrigate the crops. The following rows represent different levels of allocation of water for irrigation. The results show that when more water is available, farmers choose to plant the most productive option already in November or February (options 7 and 6 respectively). When there is less water they select to plant less maize or nothing at all (option 1). In years of water scarcity, such as 2005, we can see in the table that this is the case even if 80% of the total water is used for irrigation.

An allocation factor of 0.55 was selected for the following tests, since it is found to be a tipping point between good and poor availability for many of the years in the tested period and therefore allows for a higher range of represented situations. With this level of allocation, the area receives an amount of water that would be able to satisfy the full demand of the most productive alternative of crops in 10 out of the 14 years, with 4 years experiencing water shortages, reflecting the number of drought events in the 2000-2014 period.

### 3.3.2 Value of additional information for the decisions

The model was run with the optimised thresholds as well as with the 10 additional sets of thresholds. The optimised thresholds together with the reservoir levels and snow cover values from which they are derived are shown in Figure 4. The points are coloured according to the decisions taken with perfect information at each of the decision points. The objective of this step is to identify a set of thresholds that maximises the number of points that are correctly classified according to the optimal path. The perfect classification when using only reservoir level data would have all the years in which the good availability path should be followed (coloured in blue) above the dashed threshold and the years in which the poor availability path should be followed (coloured in red) below it. The additional dataset, if useful, should then help to improve the classification with a second threshold. In this case, the years in which the snow cover is above a certain threshold (dashed line in the lower plot)



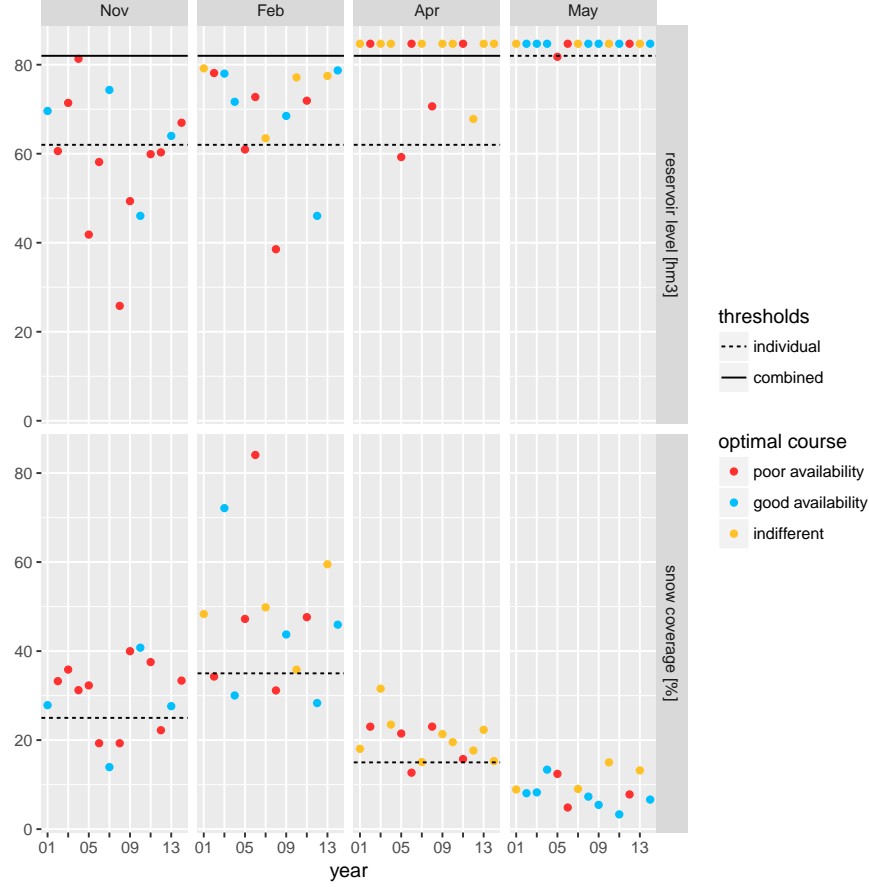

**Figure 4.** The points represent reservoir levels (upper plot) and snow coverage (lower plot) for the period 2001-2014. The points are coloured according to the decisions taken with perfect information at those decision points. The individual thresholds mark the threshold for reservoir level or snow coverage when considered independently, while the combined thresholds are the modified thresholds for reservoir level for years in which the snow coverage threshold is not reached.

are classified with the same threshold used when no additional information is considered, while the years that have snow cover below the threshold require a higher reservoir level in order to classify the availability as good. The optimised thresholds for the reservoir level were established at 62 hm$^3$ for November, February and April, and 82 hm$^3$ for May; the thresholds for snow cover were set at 25%, 35% and 15% for November, February and April respectively, while the increase in the reservoir level

5   threshold for the years that are below the snow threshold was set at 20 hm$^3$.

Figure 5 shows the total benefit obtained during the period tested using reservoir level data only and using both the reservoir level data and the snow cover data with each of the 10 sets of thresholds and with the optimised set identified in the previous step (labelled as 62 in the figure). For each threshold, two columns are shown. The column on the left provides the gains and losses when considering only the reservoir levels, while the column on the right shows the benefits and losses when both snow



cover and reservoir levels are considered. The length of the columns is determined by the accumulated years with net gains (above zero), and the accumulated years with net losses (below zero). The black dot represents the net benefit for the whole period. The gains and losses obtained using perfect information (right column) and no information are included in the first two columns as reference, and are independent of the thresholds. The columns are coloured to show the total gain or loss of

each of the years in the period. Figure 6 presents the Relative Value (RV) of the decisions using each of the two tested sources of information with respect to the decisions informed by perfect information and the uninformed decisions. This shows that the relative values for the total benefits are negative for almost all thresholds, both when using only reservoir levels as well as when also using additional information of snow cover. This means that, for the period as a whole, selecting a course of action based on the expected availability informed by these datasets does not result in higher benefits than when following the path

that performs best on average every year. The reason for this lies in the large losses incurred when failing to recognise a poor availability year and as a consequence planting more than what can be irrigated. This is the case for the years with a negative benefit represented in Figure 5. These high losses also result in higher thresholds showing a better relative value, since these thresholds lead to more years being regarded as poor availability years, thus leading to lower areas being planted.

Still, the results show that the additional information does help to reduce the losses in some of the years and for all thresholds

a better relative value is obtained when using the additional dataset on snow cover. The total benefit for the period 2001-2014 (represented by a black dot in Figure 5) is higher for all thresholds.

The high losses in some of the years are the result of the limited profit margin between the cost of planting and the selling price of the products. To illustrate further the effect of the profit margin in the decision and the value of information, we have run a series of additional simulations where the costs of planting is reduced by 50%, 75% and 100% (or zero cost). The relative

value of information for these simulations shown in Figure 7 indicates there is a gradual increase in the relative value of the informed decisions as the ratio of the benefits from the crop yield to the cost of planting increases. The fully detailed gains and losses for these simulations can be found in the supplementary material (S1). Relative values are still low, however, even when there is no cost for planting. This is because the uninformed decision used as a reference also improves with the reduction of the cost. The course of action that performs better on average, in which the uninformed decision is based, is path 3 for the full

reported cost, path 4 for the reduced costs and path 5 when no cost is considered. This means that with lower or no investment cost in terms of planting it is better on average to plant the more water demanding crops. This reduces the relative value of the informed decisions for the years in which the optimal path is followed.

At the reduced costs it also appears that the added value of the information from snow cover reduces, and in some cases is even detrimental, particularly at the higher reservoir level thresholds. This is likely caused by the uncertainty in of the

relationship between snow cover and available water resource, which will be elaborated on in the discussion.

## 4  Discussion

To answer the question posed in the title if users can benefit from additional information to support operational drought management decisions, we follow an approach that combines stakeholder consultation to be able to understand the decisions users




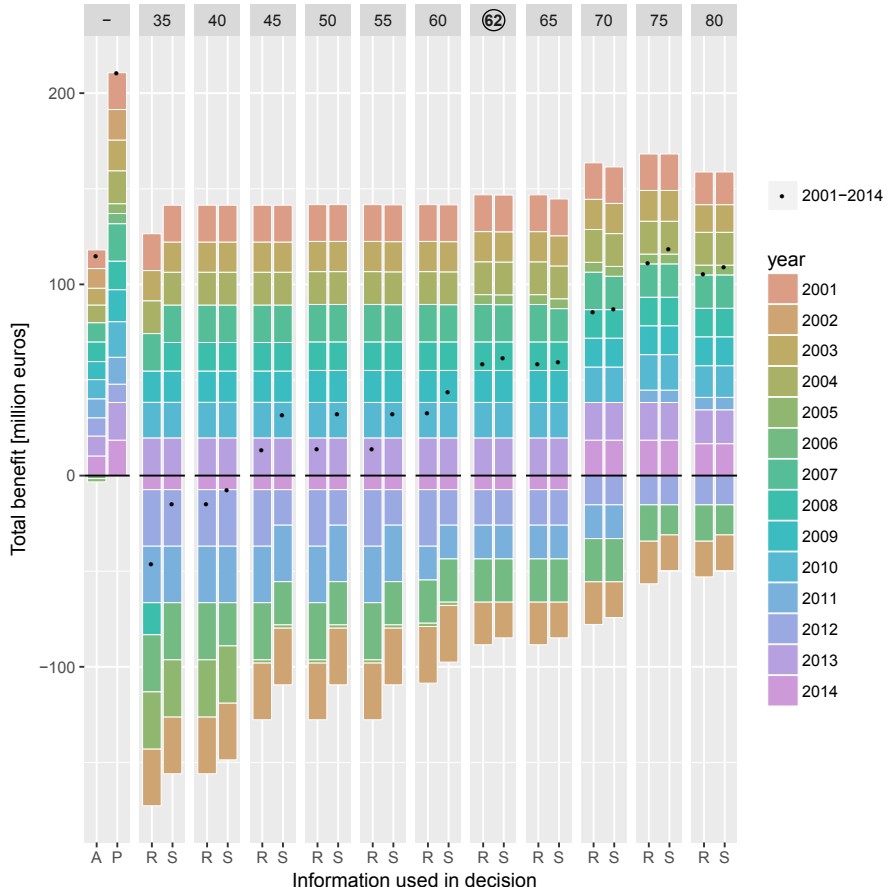

**Figure 5.** Total benefit for decisions informed by reservoir level alone (R) and with the addition of snow information (S) for the 10 sets of thresholds and the optimized thresholds (labelled as 62). The total benefit for uninformed decisions (A) and perfect information (P) is included as a reference. The colours indicate the yearly benefit while the points represent the total benefit for the period (total gains-total losses).

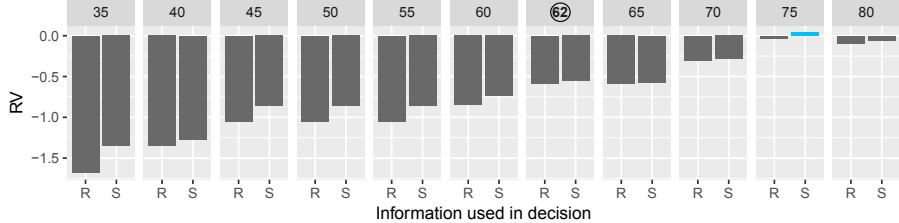

**Figure 6.** Total Relative Value for the period 2001-2014 for decisions informed by reservoir level alone (R) and with the addition of snow information (S) for the 10 sets of thresholds and the optimized thresholds (labelled as 62).





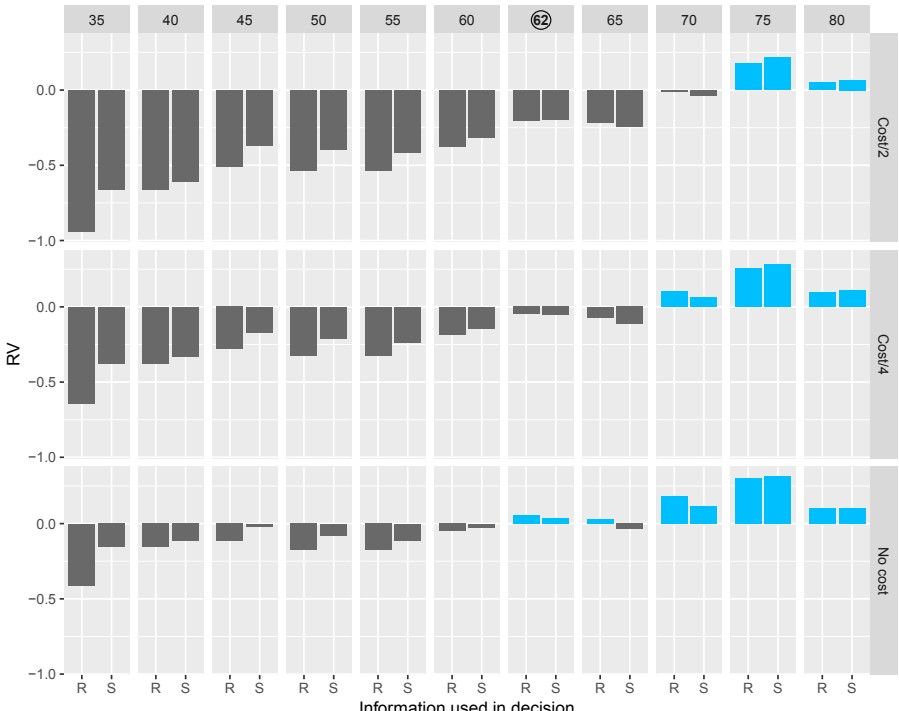

**Figure 7.** Relative value with different levels of cost for planting.

make and how they use information to support those decisions, and decision modelling to explore how additional information can be used to inform and influence the decision process. As stated by Iglesias et al. (2017), the main advantage of the semi structured interviews method is these encourage discussion, while the main limitations are related to the small sample size which means that only a partial view is obtained of the plurality of the stakeholders that make these decisions. Despite the

5 limitations, the responses of the interviews provided a detailed description of the possible choices to deal with water shortages in the Ebro basin, and the interaction and feedbacks between water management strategies at basin, irrigation district and local farmer scales by providing increased comprehension of the stakeholders' views. This knowledge was then used to explore the value of data products that the users can benefit from.

Based on these findings, a model of the interrelated decisions of farmers and water managers at the irrigation district scale

was built. The objective was to quantitatively assess the effect of information on the decisions. The decisions modelled are informed by the expected water availability during the irrigation season, which is currently derived mainly from the reservoir levels. The model can, however, be used to test any dataset that can inform the represented decisions. In this case we test the use of additional information of remotely sensed snow cover as this is information users currently may consider, but further research on the value of different datasets that inform the expectation of the available water resource could be conducted using

the model.





The snow cover product tested is a medium resolution global remote sensing product. Gascoin et al. (2015) showed the value of this product to provide snow cover information at the Pyrenees range scale. In our analysis we used the information from this product at the basin headwater scale and the results show improvements in the decisions with respect to using the reservoir levels alone. These improvements in the relative value, although small, correspond to significant reductions of losses. These

losses occur in 2002, 2006, 2011 and 2012, which match the years for which drought impacts on irrigation agriculture have been reported (Linés et al., 2017), and are the result of an inappropriate course of action being chosen as a result of the expected availability of water being too high, compounded by the high cost of planting relative to the return on investment of the crops planted. To test the robustness of the observed effect of the additional information, the model was run 10 times with random values of snow cover. The results of these runs (included in the supplementary material, S3) show that the improvement then

also follows a more random pattern and in some cases are detrimental, thus supporting the hypothesis that the improvements in the decisions are indeed caused by the additional information on snow cover.

Decisions taken with perfect or no information were used as reference cases. The difference in benefit between these two cases reveals the potential improvements that information can bring with respect to the uninformed decisions. With perfect information, losses can be avoided in seasons of water scarcity and benefits maximised when enough water is available. It

should be noted that the paths for perfect information as included in the model maximise the benefit of the whole group of farmers, rather than that of individual farmers. In reality, however, benefits and losses are not shared by the group and individual farmers would try to optimise their individual benefit instead, though community collaboration in the form of the established user associations in the basin ensure that a tragedy of the commons does not arise.

The uninformed case follows a conservative approach considering every year that water will be limited. In this way high

losses are avoided, but the benefits are well below the potential. In this scenario, information is expected to help the decision makers in characterising each season in terms of the water availability and selecting what and when to plant accordingly, increasing their benefits. However, the results of the model indicate that selecting the option that performs better on average, as it is done in the uninformed case, leads to higher benefits than using the information on reservoir levels (alone or supported by the MODIS snow cover information) to decide what the best option for that particular year is. This is in contradiction with

the current practice, in which the reservoir level information is used to support the decision and different choices are taken each year. The reason for these differences may lie in the fact that not all the losses are assumed by the farmers, since there are subsidies for certain crops or for losses incurred in disastrous years, and these are often based on planted surface, and will influence the ratio between the return from the crop yield and the investment costs incurred when planting. In addition, the actual farmer decision on what to plant is influenced not only by water availability, but also by the market prices of the crops

and these and other subsidies. Maize has a high cost of production and therefore, when its selling price is low, farmers tend to select other crops with lower production cost (Espluga Trenc, 2016). In the model, however, the planting costs and selling prices were kept constant for all the years to better observe the effect of information in the selection of the crops.

The effect of the cost of planting and the profit margins on the usefulness of the information was explored by running the model for different planting costs. Changing the cost of planting modified the course of action both for the informed and

uninformed decisions. The reduction of the costs results in higher relative values for the informed decisions for the period as a





whole caused again by the reduction of the net losses. The ratio between the cost of planting and the return on investment on the crop is similar to the cost-loss ratio used in evaluating the benefit of flood warnings (Verkade and Werner, 2011). Where the cost-loss ratio is high, the cost of taking an action in vain (false alarm) is also high, and significant losses may be incurred. This may even result in the information, which does contain uncertainty and may therefore lead to wrong choices being made,

to be detrimental. Larger losses than if that information is simply ignored and the business as usual action is taken may then be made. For users with a lower cost-loss ratio, explored here by lowering the cost of planting, additional (uncertain) information becomes increasingly valuable as these users become more tolerant to taking a wrong decision. The role of uncertainty in the link between the information used (reservoirs levels and snow) and the realisation of the available water resource is not explicitly explored in this study through for example a hydrological model, though explicitly considering the uncertainty can

add further value to the information. Several authors (Roulin, 2006; Verkade and Werner, 2011) have shown that the value of information from forecasts is always higher when these are probabilistic. In the application presented here, the relation between the reservoir levels and the available water resources is more certain than the relation between the snow cover and the available water. This may also explain the poorer performance when using snow cover information than when using only reservoir levels such as occurs in 2006. In this year the snow cover at the start of the year (February) was exceptionally high,

leading to an expectation of good water resource conditions. However, this was due to a widespread snowfall at the end of January just before the decision point in February. This snow melted rapidly and snow cover in April was anomalously low, with low water resource availability for the rest of the season.

Detailed analysis of the years where the benefit of using the additional information shows that this arises mainly from the reduction of the losses in those years in which the optimal decision to take is more uncertain. These are the years when

the classification of the water resource as being good or bad is difficult, and the added value of the additional information on the snow cover is in that it makes it more difficult for a bad year to look good. However, the value of the additional information is not equal to each of the different types of farmers identified. The additional information on the expected water resource provided by the snow cover is relevant only to the decisions that are made in February. For the technified farmers the information is therefore of little value, as the main decisions made by them fall in November and in May, respectively before

the snow accumulation period, and after the snowmelt period.

For the small scale farmers, the additional information can be relevant for the decisions that are made when there is snow cover, primarily those made in February, but also those made in April. However, the benefit is again not evenly distributed. These small scale farmers were divided into three groups of decreasing risk averseness (R1, R2 and R3). We find that the additional information benefits the group of farmers that is willing to take more risk most. These are the farmers that decide

to take the risk to wait for a possible improvement when the water availability is classified as not being good at the decision moment, instead of taking the safe bet and securing a crop by planting a barley crop, which does not depend on irrigation and possible curtailments. The most risk averse small scale farmers (R1), do not even wait for any information on water availability, and already plant barley in November. For them there is no value in the additional information. For the slightly less risk averse farmers (R2) there is limited value in the additional information. If in January the water resource situation is expected to be

good, then they will choose to plant maize, but at the first sign of it being bad they will forfeit the possible higher profits





from maize and opt to take the safe bet by planting barley. The most risk acceptant small scale farmers benefit the most from the additional information, as it will help them make the choice between taking the gamble of waiting for the water resource availability become better so that they can plant maize, or if it does not run the risk of having to leave the land fallow. These small scale farmers have only one crop, so once the decisions is made there is no further value to information.

This is an important result as it demonstrates how the value of information depends on how it can improve advanced insight into the probable state of nature, which is important to the expected utility of the decision. However, it also depends on the level of risk aversion of the user. For users that are very averse to risk, there is no added value to using the information in this case on the snow-cover as they will take the safe bet and plant barley at the first sign of poor availability in November. The added value is the highest to those small scale farmers (R3) willing to take the risk of waiting for water resources improving

before taking action. In this case these results show that the additional information may be beneficial to improved equity across the farmers in the irrigation district as it is most beneficial to small-scale farmers, provided they are willing to take a gamble to improve their benefits. In this paper we model the distribution of risk averseness using only a simple percentile distribution. A more realistic distribution of risk averseness can be developed using for example the Constant Absolute Risk Aversion Utility Function (Matte et al., 2017; Quiroga et al., 2011), though this will require extensive survey data to determine how risk

averseness is distributed among farmers.

## 5   Conclusions

Operational drought management decisions in the Ebro basin were examined with the aim to assess the role of information in these decisions, following an approach that combines stakeholder consultation and decision modelling, allowing a comprehensive analysis of the role of information on drought management decisions in the area. Consultation with the different

decision makers in the Ebro basin provided useful insight into the operational decisions they take in managing water resources when scarce, and their information needs and use. This allowed us to identify the courses of action available to the farmers and water managers, and to analyse their choices as a function of the information they have available to them. Feedbacks between the decisions made by farmers and the reservoir operators at irrigation district level were identified: Curtailments imposed at irrigation district level as a result of water scarcity influences the decision farmers make on the planting of crops, which in turn

influence demand and consequently water scarcity.

Based on the findings of the consultation, a decision model representing these interrelated decisions was built with the aim to quantify the effect of additional information on the decisions. The modelled decisions, which consider the allocation of water, are taken based on the expected availability of water during the irrigation season. This is currently informed primarily by observed reservoir level data. When levels are above a defined threshold at the time of the decision, water resources availability

is classified as good, whereas when levels are below the threshold and expected demand is high it is classified as poor and curtailments to water allocations applied. Farmers decide on the crop to be planted based on their expectation of water resources availability, and whether curtailments are in force. The decision model was then extended from considering only reservoir levels



to include additional information on snow cover in the basin headwaters obtained from MODIS remote sensing data to inform the expectation of water resources availability.

Our simulations with the decision model show the additional information can contribute to better decisions and ultimately to higher benefits for the farmers. However, the ratio between the cost of planting and the market value of the crop proved to be

a critical aspect in determining the best course of action to be taken and the value of the (additional) information. When there is little room for error due to small margins, then any information used to inform the decision may even be detrimental to any benefits being made. However, even in this case the additional information on snow cover can provide benefit over using the reservoir levels alone. Tests with reduced planting costs, thus increasing margins, does lead to additional benefit when using the additional information from snow cover, although uncertainty in the relationship between good snow cover and water resource

availability may lead to overestimation of the expected resource, and consequent losses.

A key finding of our research is that farmers can benefit when the operational decisions they make consider the additional information. To what extent they benefit does, however, depend to a great extent to their level of risk averseness. Risk-averse farmers will decide to take the safe option early on, with information on the available water resource then having no value. Farmers that are less risk averse do benefit as the information helps them weigh the options between planting a crop with a

higher return, or having to leave the land fallow.

*Code and data availability.*  All in situ and remote sensing data used in this research are openly available. The sources are mentioned in Sect. 2.3. The crop models, Aquacrop-OS and Cropwat, are open source and available from http://aquacropos.com/ and http://www.fao.org/land-water/databases-and-software/cropwat/en/ respectively. The code for the decision model may be made available by request to the corresponding author.

*Competing interests.*  The authors declare that they have no conflict of interest.

*Acknowledgements.*  The authors would like to thank CHE, CAyC and Raimat for their collaboration in the consultation phase of this research. This research received funding from the European Union Seventh Framework Programme (FP7/2007–2013) under grant agreement no. 603608, "Global Earth Observation for integrated water resource assessment": eartH2Observe. This work is a contribution to the Hymex Drought and Water Resources Science Team.




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





Gil Martínez, M.: Cebada y maíz rastrojero. Productividad económico-ambiental de la fertilización con purín (Barley and maize. Economic and environmental productivity of fertilisation with liquid manure), Informaciones técnicas 245, Gobierno de Aragón. Departamento de Agricultura, Ganadería y Medio Ambiente. Dirección General de Alimentación y Fomento Agroalimentario. Servicio de Recursos Agrícolas, Zaragoza (Spain), 2013.

Gutiérrez López, M.: Orientaciones varietales para las siembras de cereales en Aragón. Resultado de los ensayos. Cosecha 2011 (Guidance on crop varieties for cereal cultivation in Aragón. Trial results. Season 2011), Informaciones Técnicas 230, Diputación General de Aragón. Departamento de Agricultura y Alimentación. Dirección General de Desarrollo Rural, Servicio de Programas Rurales, Zaragoza (Spain), 2011.

Gutiérrez López, M.: Transferencia de resultados de la red de ensayos de maíz y girasol en Aragón. Campaña 2015. (Transfer of results
from the maize and sunflower trial network in Aragón. Season 2015), Informaciones Técnicas 259, Gobierno de Aragón. Departamento de Desarrollo Rural y Sostenibilidad. Dirección General de Desarrollo Rural. Servicio de Innovación y Transferenca Agroalimentaria, Zaragoza (Spain), 2016.

Hall, D., Salomonson, V., and Riggs, G.: MODIS/Terra Snow Cover 8-Day L3 Global 500m Grid, Version 5 [h17-18v04], http://dx.doi.org/10.5067/{C}574UGKQQU1T, 2006.

Harrell, M. C. and Bradley, M. A.: Data collection methods. Semi-structured Interviews and focus groups, Tech. rep., RAND, Santa Monica, CA (US), 2009.

Iglesias, A. and Garrote, L.: Adaptation strategies for agricultural water management under climate change in Europe, Agricultural Water Management, 155, 113–124, https://doi.org/http://dx.doi.org/10.1016/j.agwat.2015.03.014, 2015.

Iglesias, A., Sánchez, B., Garrote, L., and López, I.: Towards Adaptation to Climate Change: Water for Rice in the Coastal Wetlands of
Doñana, Southern Spain, Water Resources Management, 31, 629–653, https://doi.org/10.1007/s11269-015-0995-x, 2017.

Linés, C., Werner, M., and Bastiaanssen, W.: The predictability of reported drought events and impacts in the Ebro Basin using six different remote sensing data sets, Hydrol. Earth Syst. Sci., 21, 4747–4765, 2017.

Lloveras, J., Martínez, E., and Santiveri, P.: Influencia de la fecha de siembra en el maíz en los regadíos del Valle del Ebro (Influence planting date on maize crops in the irrigated areas of the Ebro Valley), Tech. Rep. 36, Agrotecnio. Universidad de Lleida, Lleida (Spain), 2014.

Macauley, M. K.: The value of information: Measuring the contribution of space-derived earth science data to resource management, Space Policy, 22, 274–282, https://doi.org/10.1016/j.spacepol.2006.08.003, 2006.

MAGRAMA: Resultados técnico-económicos de Cultivos Herbáceos 2014 (Tecnichal-economical results of herbaceous crops 2014), Tech. rep., Subdirección General de Análisis, Prospectiva y Coordinación, Subsecretaría. Ministerio de Agricultura, Alimentación y Medio Ambiente, Spain, 2015.

Matte, S., Boucher, M.-A., Boucher, V., and Thomas-Charles, F. F.: Moving beyond the cost–loss ratio: economic assessment of streamflow forecasts for a risk-averse decision maker, Hydrology and Earth System Sciences, 21, 2967, 2017.

Mylne, K. R.: Decision-making from probability forecasts based on forecast value, Meteorological Applications, 9, 307–315, 2002.

Onoda, M. and Young, O. R., eds.: Satellite Earth Observations and their impact on society and policy, Springer Open, Singapore, 2017.

Quintilla, R., Portero, C., and Casterad, M.: Apoyo a la gestión del agua en alta en la zona regable del Canal de Aragón y Cataluña con
teledetección (Remote sensing support to reservoir water management in Canal de Aragón y Cataluña irrigated area), in: Proceedings of the XXXII Congreso Nacional de Riegos, p. 9, Madrid, 2014.



Quiroga, S., Garrote, L., Iglesias, A., Fernández-Haddad, Z., Schlickenrieder, J., de Lama, B., Mosso, C., and Sánchez-Arcilla, A.: The economic value of drought information for water management under climate change: a case study in the Ebro basin, Natural Hazards and Earth System Sciences, 11, 1–15, 2011.

R Core Team: R: A language and environment for statistical computing, https://www.R-project.org/., 2016.

Roulin, E.: Skill and relative economic value of medium-range hydrological ensemble predictions, Hydrology and Earth System Sciences Discussions, 3, 1369–1406, 2006.

Stanski, H. R., Wilson, L. J., and Burrows, W. R.: Survey of common verification methods in meteorology, Research report MSRB 89-5, Atmospheric Environment Service, Downsview, Ontario (Canada), 1989.

Verkade, J. and Werner, M.: Estimating the benefits of single value and probability forecasting for flood warning, Hydrology and Earth
System Sciences, 15, 3751–3765, https://doi.org/10.5194/hess-15-3751-2011, 2011.

Wilks, D. S. and Murphy, A. H.: A decision-analytic study of the joint value of seasonal precipitation and temperature forecasts in a choice-of-crop-problem, Atmosphere-Ocean, 24, 353–368, https://doi.org/10.1080/07055900.1986.9649257, 1986.

Williamson, R. A., Hertzfeld, H. R., Cordes, J., and Logsdon, J. M.: The socioeconomic benefits of Earth science and applications research: reducing the risks and costs of natural disasters in the USA, Space Policy, 18, 57–65, 2002.

Zhu, Y., Toth, Z., Wobus, R., Richardson, D., and Mylne, K.: The economic value of ensemble-based weather forecasts, Bulletin of the American Meteorological Society, 83, 73–83, 2002.