# Peer review of "Do users benefit from additional information in support of operational drought management decisions in the Ebro basin?"

_Hydrology and Earth System Sciences, 2018_

## Referee Comment (RC1) · Anonymous Referee #1 · 25 Jun 2018

This paper studies the value of extra information regarding water availability for drought management decisions, focusing on reservoir managers and irrigators. This is an interesting topic and the outputs of the work are relevant to the scientific community. However, there are a few issues in the current version that need to be explained further so the manuscript can be ready for publication.

In Section 2.1, when describing your study area, it would be nice to add something related to how droughts have impacted this basin in the past and what were the implications for agriculture in the area.

In Section 2.2.1, there is hardly any information about how many people do they

interview from each group (reservoir operators, irrigators), what type of question-naire/survey method was used (you mentioned it briefly in the discussion but I think it should be explained here in more detail), etc. I think this is needed to understand a bit better how the assumptions you are making in your model are representing the sector. Authors acknowledged that doing a larger survey would be ideal although out of the scope of the work. But still I think readers need to know more about the survey process. Also, they stated that irrigators are very different in their risk aversion levels and this will affect their decisions during a drought. Without information of the sample, readers might wonder if the participants in the survey are representing that spread.

In page 13, line 4, you said that you explore different thresholds "keeping the same thresholds at each of the four points where the farmers make decisions during the season". However, in page 16, you give different thresholds for Nov, Feb, April and May. Could you clarify this?

In Table 3, is there any way authors can add some kind of information to show how dry or wet each of the years under study was. From what I understood, 05 and 06 will be drier years as even with high allocation factors the result is not very good...But maybe adding some information (e.g., a standardized precipitation index) could help to interpret the results.

Figure 4, after looking at it a few times, is still a bit confusing to me. Authors could review the explanation of it and/or the presentation of it to make it easier to follow.
* * *

---

## Referee Comment (RC2) · Anonymous Referee #2 · 23 Jul 2018

I have approached "Do users benefit from additional information in support of operational drought management decisions in the Ebro basin?" by Clara Linés and co few times by now. The paper touches a very timely topic and (it seems that it) takes an interesting approach to quantify the value of different information attributes on decisions that various stakeholder can take across various spatial scales. The paper demonstrates application of the methodology is Ebro River Basin in northern Spain, which is highly regulated and intervened by socio-economic activities, in particular irrigated agriculture and hydropower generation. As far as the context of the paper is concerned, the paper is certainly inline with aims and scope of HESS and can attract a large portion of the journal's readership. However, at this stage the paper suffers from major

issues. In particular, the paper is quite disorganized in terms of sectioning and the sequence of materials provided. Second, it seems that presentation in the paper lacks effective strategy, which hinders the reader to get involved with the paper. Finally, the level of details regarding the data obtained through interviews, methodology used for modeling, experimental setup and investigation made is quite low, in a way that the work is indeed not producible, if some one wants to apply the same approach in another case study. I do believe the paper should go under major revisions, in terms of the rationale and the content provided and resubmitted for another round of review, this time focusing on the specific results and findings.

Below, please see my specific comments

1) The paper is poorly written. The use of English can (and must) be improved in many parts of the paper (e.g. P1, line 8; P2, line 7-9 among others). In addition, it is very hard to read the whole paper in one sitting (at least I was not able to accomplish) due to long sentences and the existence of a lot of text. I strongly suggest a major editorial effort before the paper resubmitted.

2) It seems that the paper has missed positioning itself in the broader context of the current socio-hydrology research. On the one hand, review of previous studies in other parts of the world has been largely ignored. This includes for instance missing previous works on performing semi-structured interviews, developing decision support models through stakeholder engagement, and quantifying the value of information. The paper requires framing itself very clearly in the introduction.

3) The paper is extremely disorganized and is poorly sectioned. The section related to Results in particular is very long relative to other sections and is hard to follow. Most importantly, the results section includes even the results of semi-structured interviews that basically provides the data support for developing the decision model. I believe a great portion of what is presented in the results can go under a new section related to the data support and model development.

4) The way that paper presents the information and findings through text, tables and figures seems not very well thought. For example, a central part of this paper is the decision model, with which the value of new information can be assessed for various stakeholders. After reading the paper this part of the paper quite a few times, I am not still slightly clear about how the model has been developed. A schematic and some formulas would certainly help. To facilitate following the paper, I believe a standalone section is required to discuss the experimental setup and how the results should be viewed. Figures are very hard to understand. Similar to the other reviewer, I do also have problem with understanding Figures 4 (and 5 and 6 and 7). The discussion is also extremely long and rather scrambled. I believe synthesizing information under appropriate subsections would be very helpful.

5) While the paper is long, it does not provide information required to reproduce the work or to at least understand the process of data gathering through semi-structured interviews and model development. As noted above, it is not clear how the model has been developed as a result it is not possible to really examine the truthfulness of the results and the relevance of the discussion provided.

---

## Author Response (AR1)

**Content**

…………………………………………………………………………………………………………………………………………..

**Point-by-point response to the reviews**

Reply to Anonymous Referee #1

We thank the reviewer for taking the time to review the manuscript and for the helpful comments and suggestions. Here we provide answers to the specific comments and indications of how the manuscript could be improved to address the issues raised by the reviewer.

………………………………………………………………………………………………………………………………………

**This paper studies the value of extra information regarding water availability for drought management decisions, focusing on reservoir managers and irrigators. This is an interesting topic and the outputs of the work are relevant to the scientific community. However, there are a few issues in the current version that need to be explained further so the manuscript can be ready for publication.**

**1) In Section 2.1, when describing your study area, it would be nice to add something related to how droughts have impacted this basin in the past and what were the implications for agriculture in the area.**

Three drought events that produced impacts on the agriculture and other sectors have been recorded in the area for the period 2000-2014: a short drought spell in 2002, a multiyear event that lasted from the winter of 2004-2005 to the spring of 2008, and another during the years 2011 and 2012 (Linés et al, 2017). The impacts of the multiyear drought of 2004-2008 in the Ebro basin have been widely studied. The north-eastern part of the basin was the most impacted (Hernández-Mora et al, 2013). Agriculture was the most affected sector, with 540 million of euros of estimated losses in crop production during the hydrological year 2004-2005 and further losses of 272 million in related industries (Perez y Perez & Barreiro-Hurlé, 2009).

Suggested change: We will include this information in the study area section.

**References:**

Hernández-Mora, N., Gil, M., Garrido, A., & Rodríguez-Casado, R. (2013). *La sequía 2005-2008 en la cuenca del Ebro. Vulnerabilidad, Impactos y Medidas de Gestión* (p. 62). Universidad Politécnica de Madrid - Centro de Estudios e Investigación para la Gestión de Riesgos Agrarios y Medioambientales - CEIGRAM.

Linés, C., Werner, M., & Bastiaanssen, W. (2017). The predictability of reported drought events and impacts in the Ebro Basin using six different remote sensing data sets. *Hydrol. Earth Syst. Sci.*, *21*, 4747–4765.

Perez y Perez, L., & Barreiro-Hurlé, J. (2009). Assessing the socio-economic impacts of drought in the Ebro River Basin. *Spanish Journal of Agricultural Research*, *7*(2), 269–280.

**2) In Section 2.2.1, there is hardly any information about how many people do they interview from each group (reservoir operators, irrigators), what type of questionnaire/survey method was used (you mentioned it briefly in the discussion but I think it should be explained here in more detail), etc. I think this is needed to understand a bit better how the assumptions you are making in your model are representing the sector. Authors acknowledged that doing a larger survey would be ideal although out of the scope of the work. But still I think readers need to know more about the survey process. Also, they stated that irrigators are very different in their risk aversion levels and this will affect their decisions during a drought. Without information of the sample, readers might wonder if the participants in the survey are representing that spread.**

One interview session was held at each of the locations (the basin authority, the irrigation association and the farm) with two or three people participating in each of them. In the interview that was held at the basin authority, the participants included the head of one of the basin's management units and two members of the hydrological planning office, both with expertise in drought management in the basin. In the interview at the Irrigation Association, the participants were the head of the Irrigation Association and the engineer in charge of the information service about current and expected water availability. And in the interview at the farm level, two people participated: the head of viticulture and engineer responsible of the information service.

The method used was semi-structured interviews. We had a prepared set of questions to ensure we collected all the details that we needed about their decision processes, the data they use to inform them and the gaps or limitations they identify in the information available to them. These questions were used during the interview as a guide and checklist of the topics of interest that we wanted to cover. As is commonplace in semi-structured interviews, we asked the participants to tell us about their own practices, as well as the practices of the groups they deal with in relation with drought management, letting them tell the story in their own way. We only used the questions to bring out the topics that were not mentioned by the participants to ensure we covered all the topics we intended to cover.

As indicated in section 3.1.3, the information about the practices and attitudes of farmers in the study area was to a great extent obtained from the interview with the Irrigator Association who works in close collaboration with all the farmers in the area and has therefore a wider view than individual farmers. We acknowledge that the sample of interviews is held among a small sample. The objective of these interviews was primarily to build our understanding of how the farmers behave as a function of their expectation of water availability. Although important to the assumptions made in the development of the decision model that we constructed, we would readily agree that our representation of the farmer behavior and the

diversity of responses across farmers may be over simplified. A more complete interview would reveal more information, but we feel that is outside the scope of this paper.

Suggested change: We will add this additional detail on the interview of the participants and method to the section 2.2.1. In the revised version we also propose to put less emphasis on the interviews as we acknowledge that the sample of interview is small, and that the representation of the diversity of farmer responses may be simplified.

**3) In page 13, line 4, you said that you explore different thresholds "keeping the same thresholds at each of the four points where the farmers make decisions during the season". However, in page 16, you give different thresholds for Nov, Feb, April and May. Could you clarify this?**

In page 13, line 4. We apologise for the confusion. For each of the four decision moments (Nov, Feb, Apr, May) different thresholds are used, but we meant that the same set of 4 thresholds is kept for all the years analysed.

Suggested change: We agree that is confusing and we will rephrase the sentence in page 13, line 4, to make it clearer.

**4) In Table 3, is there any way authors can add some kind of information to show how dry or wet each of the years under study was. From what I understood, 05 and 06 will be drier years as even with high allocation factors the result is not very good...But maybe adding some information (e.g., a standardized precipitation index) could help to interpret the results.**

Yes, in 2005 and 2006 there was a drought event, as well as in 2002 and 2012 (see answer to comment 1). This is indeed well represented by the SPI data of the study period (Fig C4).

[Figure]

Fig. C4. Monthly SPI-12 (upper plot) and SPI-12 for the month of September (lower plot) for the catchment area of Barasona and Santa Ana reservoirs. Calculated with CHIRPS precipitation data for the period 1981-2015.

Suggested change: We will include the timeline for SPI-12 for September information in the header of table 3 of the manuscript to provide an indication of the dryness of each year.

**5) Figure 4, after looking at it a few times, is still a bit confusing to me. Authors could review the explanation of it and/or the presentation of it to make it easier to follow.**

We agree that Figure 4 is complex and contains a lot of information. This figure aims to illustrate the optimised thresholds. A threshold is required when taking the decision to choose between two options as a function of a certain indicator. In this case the threshold is the value above which the farmer would consider that the availability of water is good enough to plant the most productive crops (this corresponds to following the paths marked with a blue A in Figure 3). The threshold is perfect if it always makes the farmer take the decision that results in a higher benefit at the end of the season. Using perfect information, we first identified the decisions that result in the highest benefits to the farmers. These decisions are represented in Figure 4 by the coloured points, with the level of the reservoir on the y-axis. If the point is red, then it is best for the farmer to follow the path marked with a red A in Figure 3, while if it is blue then it is best to follow the blue A path. If it is yellow, then it does not matter which of the paths is followed. A perfect threshold for the reservoir level to divide between good and poor water availability would be selected such that all the red points are below and all the blue points above the threshold. As can be seen in Figure 4, it is not possible to select a perfect classification of the reservoir levels with a single threshold. The dashed lines mark the thresholds that maximises the points correctly classified. The fact that the classification cannot be perfect means that the reservoir level alone does not provide enough information on what is the best decision to take. Additional information would be valuable if it contributes to improve the classification and, therefore, results in the decision that maximises the benefits being taken more often. In this case the additional information we consider is the snow coverage data. We show this in the lower part of the figure. Again the figure shows that this information alone does not lead to a perfect classification either (dashed lines).

This is different when the combined information of reservoir level and snow cover extent is used to inform the expectation of water availability. We incorporate this additional information by amending the threshold of the reservoir level. This is the solid line in the figure for the Months Nov, Feb and Apr. Snow cover is not considered in May as that is too late in the season to be of significance.

When snow covers an area larger than the threshold coverage (dashed line in the lower plot), then the original threshold for the reservoir level is used. However, when the snow coverage is smaller than the identified threshold, and therefore the future contribution to the reservoir level from snowmelt is expected to be lower, a second, more conservative threshold for the reservoir level is used. With the second threshold some of the red points incorrectly classified above the original threshold are now classified below the thresholds.

Suggested changes:

- We propose to improve the explanation of figure 4 in the manuscript, describing this step by step to ensure the reader does not get confused.
- We will include the missing label in y axis ("reservoir level") in Figure 4
- We will mention in the caption of Figure 4 that the colours marked in the legend are considered as the "optimal course" and refer to the paths illustrated in Figure 3.

Reply to Anonymous Referee #2

We thank the reviewer for the time and effort to review the manuscript and for providing suggestions to improve it. We include below answers to the specific comments and indications of how the manuscript could be change to address the issues raised by the reviewer.

………………………………………………………………………………………………….

**[Referee] I have approached "Do users benefit from additional information in support of operational drought management decisions in the Ebro basin?" by Clara Linés and co few times by now. The paper touches a very timely topic and (it seems that it) takes an interesting approach to quantify the value of different information attributes on decisions that various stakeholder can take across various spatial scales. The paper demonstrates application of the methodology is Ebro River Basin in northern Spain, which is highly regulated and intervened by socio-economic activities, in particular irrigated agriculture and hydropower generation. As far as the context of the paper is concerned, the paper is certainly inline with aims and scope of HESS and can attract a large portion of the journal's readership. However, at this stage the paper suffers from major issues. In particular, the paper is quite disorganized in terms of sectioning and the sequence of materials provided. Second, it seems that presentation in the paper lacks effective strategy, which hinders the reader to get involved with the paper. Finally, the level of details regarding the data obtained through interviews, methodology used for modeling, experimental setup and investigation made is quite low, in a way that the work is indeed not producible, if some0one wants to apply the same approach in another case study. I do believe the paper should go under major revisions, in terms of the rationale and the content provided and resubmitted for another round of review, this time focusing on the specific results and findings.**

Reply: We thank the reviewer for underlining the timeliness of the topic and that its contribution to the readership of HESS. We do apologise though that it appears that the organisation of the paper was not, to the reviewer, as clear as it could be and will endeavour to improve this clarity. We hope and firmly believe that the improvements we propose to the structure will benefit its clarity and remove the confusion.

**Below, please see my specific comments**

**1) The paper is poorly written. The use of English can (and must) be improved in many parts of the paper (e.g. P1, line 8; P2, line 7-9 among others). In addition, it is very hard to read the whole paper in one sitting (at least I was not able to accomplish) due to long sentences and the existence of a lot of text. I strongly suggest a major editorial effort before the paper resubmitted.**

Reply: We have carefully analysed the sentences referred to, and can unfortunately not see any grammatical issues with these. However, we do agree that those and other sentences in the manuscript are long, and that such long sentences make reading the text more difficult.

Suggested change: We will review the manuscript and shorten sentences that are unnecessarily long, and summarise the text where possible. Also, the changes of the paper structure suggested as a response to comment 3 will help reduce the length of the paper. We will carefully revise the paper again to ensure correctness of the English grammar used.

**2) It seems that the paper has missed positioning itself in the broader context of the current socio-hydrology research. On the one hand, review of previous studies in other parts of the world has been largely ignored. This includes for instance missing previous works on performing semi-structured interviews, developing decision support models through stakeholder engagement, and quantifying the value of information. The paper requires framing itself very clearly in the introduction.**

Reply: The intended scope of our paper is to perform a stakeholder oriented analysis of the value of information for drought management decisions. We therefore positioned the paper primarily in the context of research that explores the value of information to decision making. This is the reason why we motivate in the introduction our study from that perspective, primarily providing references that call for this type of research and calling on examples of previous studies that have addressed the quantification of the value of information in support of water management decisions, both from a stakeholder oriented perspective and from a scientific perspective. We do agree that this paper also fits to some extent within the context of the developing field of socio-hydrology in that it explores the co-evolution of the availability of water and the decisions taken by humans (in this case farmer and irrigation operators). While the emerging field of socio-hydrology (Sivapalan, 2002) is a broad field (see also the review of the first biennial of the proclaimed IASH Panta Rhei decade, McMillan et al.,2016), our work is related most to the working group on Drought in the Anthropocene (van Loon et al., 2016), which explicitly addresses the inefficiency of drought management due to poorly understood feedback between people (and the decisions they make) and drought conditions.

As noted in response to the comments of the first reviewer, the objective of the semi-structured interviews was primarily to build our understanding of how the farmers behave as a function of their expectation of water availability, and we do not consider our approach to these interviews as the main contribution of this paper. We have selected this approach as semi-structured interviews are a specific method suited to gather stakeholder input. We agree that its use in hydrological research can be better contextualised by including references to other research that use semi-structured interviews to develop an understanding of stakeholder behaviour.

Suggested changes:

We will extend the introduction to position our research within the context of the emerging field of socio-hydrology, referring in particular to the current poor understanding of how the decisions people make contribute to an efficient management of drought. We will include references (see above) as appropriate. In the methodology section we add the following references about the use of semi-structured interviews:

- Carr et al (2011), that use semi-structured interviews to understand what drives the decision of farmers to reuse wastewater in Jordan.

- O'Keeffe et al (2016), that describe an example of application of semi-structured interviews to gather data on farmer water use practices in two Indian districts.

Reply: We opted to include all results, both from the semi-structured interviews as well as from the model phase, in the result section. Since the model development depends on the results of the survey phase, the drawback of this choice is indeed that a large part of the content of the manuscript needs to be in the results section.

As proposed by the reviewer, an alternative to avoid this would be to include a separate section reporting the results of the semi-structured interviews (including both method and outcomes), followed by another section that discusses the model design and options. In this case the result section would contain only the results of the model runs to quantify the value of information. This alternative structure would also shorten the text by merging the content now divided between the methods and the results section.

Suggested changes: We propose to restructure the article as suggested. The outline for the manuscript following the alternative option mentioned above is included at the end of this document (Annex 1).

**4) The way that paper presents the information and findings through text, tables and figures seems not very well thought. For example, a central part of this paper is the decision model, with which the value of new information can be assessed for various stakeholders. After reading the paper this part of the paper quite a few times, I am not still slightly clear about how the model has been developed. A schematic and some formulas would certainly help. To facilitate following the paper, I believe a standalone section is required to discuss the experimental setup and how the results should be viewed. Figures are very hard to understand. Similar to the other reviewer, I do also have problem with understanding Figures 4 (and 5 and 6 and 7). The discussion is also extremely long and rather scrambled. I believe synthesizing information under appropriate subsections would be very helpful.**

Reply: The model combines the decision of the farmers of what and when to plant and the decision of the reservoir operators on whether to apply curtailments to the amount of water that can be supplied to farmers. The choices of the farmers are schematically represented in figure 3. The information that drives those decisions and the relations between the parameters as defined in the model are represented in Table 1. We agree that the table can be difficult to follow. In figure 1 (below) we have incorporated the information from Table 1 in a schematic form to make it more visual. We will replace Table 1 with this figure and provide a succinct description in the text to explain the figure.

We also expect that including a standalone section of the model design as also proposed in the answer to comment 3 will help to make it clearer as well. A description of the new outline will be included in the "Approach and data" section (see reply to comment 3 and Annex 1).

Please, see the reply to comment 4 of reviewer 1 for an explanation of figure 4 and the corresponding suggested improvements.

The expected final results of the analysis is the total benefit for the farmers. This benefit depends on the decisions taken during each season, which in turn depend of the information used to inform them. Two information scenarios are tested: one that represents the current use of information (the decisions are informed by the reservoir levels only) and another that represents the use of additional information (the decisions are informed by reservoir levels and snow cover data). *Perfect information* and *No information* scenarios are also used as a reference of the potential value of the information. Figures 5-7 present these results. In Figure 5 the total benefit for the period for each of the scenarios and thresholds is shown. The difference between the perfect information and the no information scenarios shows the potential value of using information, as the use of uncertain information is expected to scale between these two extreme situations. This is shown in the first two columns of the Figure 5. However, as can be seen in the following columns that represent the information scenarios, the use of non-perfect information in this case can result in losses when water availability is overestimated. Figures 6 and 7 show the total relative value of the two information scenarios with respect to the reference scenarios. The relative value is negative, because the losses are higher in these two information scenarios than in the No information scenario, as seen in Figure 5. However, we can see that the losses are lower when using the additional information through the comparison of each pair of columns. The columns on the left show the relative losses when considering only the reservoir levels, while those on the right show the relative losses when also considering the snow cover. These show marginally less losses. Figure 7 has the same structure as Figure 6, and shows the result of lowering the cost of planting, and therefore reducing the losses incurred when the water availability is overestimated. As a result of the reduction in the losses, we can see that the relative value of the information increases with the reduction of the costs.

[Figure]

Figure 1. Model decisions and their inputs and outputs.

Suggested changes:
- We will replace Table 1 by Figure 1 (included below), providing also a succinct description of the figure in the text.
- We will change the outline of the paper as outlined in Annex 1 below. The expected results will be explained in the approach section to facilitate following the paper.
- The discussion section will be structured in three subsections to make this clearer (see annex 1)
- We will critically review and improve the supporting texts for figures 5, 6 & 7.

**5) While the paper is long, it does not provide information required to reproduce the work or to at least understand the process of data gathering through semi-structured interviews and model development. As noted above, it is not clear how the model has been developed as a result it is not possible to really examine the truthfulness of the results and the relevance of the discussion provided.**

Please see our replies to comment 4 above and to comment 2 of reviewer 1 for an explanation about the model development and the data gathering through semi-structured interviews respectively, and the corresponding suggested improvements to the manuscript. In addition to the explanations included in the manuscript, we are happy to share the model code upon request for reproducibility.

…………………………………………….

**Annex 1 - Alternative outline for the manuscript:**

1. Introduction

2. Approach and study area

3. Stakeholders' consultation

 3.1. Method

 3.2. Responses

  3.2.1 Confederación hidrográfica del Ebro

  3.2.2 Canal de Aragón y Cataluña (CAyC)

  3.2.3 Farmers in the Canal de Aragón y Cataluña irrigated area

4. Decision model

 4.1 Farmer Decision: Crop areas

 4.2 Reservoir operation decision: Water restrictions

 4.3 Crop water demand and benefit

 4.4 Model Options

5. Quantifying the effect of additional information

6. Results

 6.1 Farmer decisions and curtailments using perfect information

 6.2 Value of additional information for the decisions

7. Discussion

 7.1. Potential value of additional information

 7.2. Effect of the cost of planting on the value of information

 7.3. Value of the information for the different types of farmers

**List relevant of changes**

**0. General**

The manuscript has been restructured to include a separate section reporting the results of the semi-structured interviews (including both method and outcomes), followed by another section that discusses the model design and options. Previously this content was in the results section. The text has been edited accordingly and the outline suggested in the response to the reviewer has been slightly adjusted in the process [R2 C3].

We have reviewed and edited the manuscript to shorten unnecessarily long sentences and improve the readability [R2 C1].

**1. Introduction**

An additional paragraph has been added at the end of the introduction to position our research within the context of the field of socio-hydrology [R2 C2].

**2. Study area and approach**

**2.1. The Ebro basin**

An additional paragraph has been added to describe how drought has impacted the basin in the past [R1 C1].

**2.2. Approach**

An explanation of the outline of the article has been added, including the expected results, to facilitate following the paper [R2 C4].

**3. Stakeholders consultation**

**3.1. Method**

Additional details on how the interviews were conducted have been added, while the existing text has been shortened to put less emphasis on this part of the study [R1 C2, R2 C5].

Two references about the use of semi-structured interviews have been added [R2 C2]

**4. Decision model**

**4.2. Reservoir operation decision: Water restrictions**

Table 1 has been replaced by a figure (Figure 4) and the text has been edited to better describe it [R2 C4, R2 C5].

**5. Quantifying the effect of additional information**

**5.2. Model options**

Availability thresholds: The text has been edited to make this section clearer [R1 C3].

Allocation table: The header of table (now table 2) has been updated to provide an indication of the dryness of the years by including the SPI-12 value for the month of September for each of the years [R1 C4].

**6. Results**

**6.1. Selection of optimal thresholds**

The text has been rewritten to improve the explanation of the process to select the optimal threshold and the description of the corresponding figure (now Figure 5, Figure 4 in the previous version of the manuscript) [R1 C5: R2 C4].

The missing y-axis label ("reservoir volume") has been added to Figure 5 [R1 C5].

The caption has been edited to mention that the colours marked in the legend are considered as the "optimal course" and refer to the paths illustrated in Figure 3 [R1 C5].

**6.2. Value of additional information for the decisions**

The text has been edited to improve the description of the figures [R2 C4].

**6.3. Quantifying the effect of additional information**

The text has been edited to improve the description of the figure [R2 C4].

**7. Discussion**

The discussion section has been divided in three subsections to make it clearer [R2 C4].

[revised manuscript text omitted]
* (pp. 571–580). Zaragoza: Universidad de Zaragoza-AGE. Retrieved from http://congresoage.unizar.es/eBook/trabajos/060_Casterad%20Seral.pdf

CHE. (n.d.). Portal de CHEbro. Retrieved 15 March 2018, from http://www.chebro.es/contenido.visualizar.do?idContenido=37945&idMenu=2167

Eden, C., & Ackermann, F. (1998). *Making strategy: the journey of strategy management*. London: SAGE Publications Ltd.

Espluga Trenc, J. (2016). What is happening with transgenic maize? (¿Qué está pasando con el cultivo de maíz transgénico?). *Soberanía alimentaria*, *26*, 34–38.

Famiglietti, J. S., Cazenave, A., Eicker, A., Reager, J. T., Rodell, M., Velicogna, I., … Zhulidov, A. V. (2015). Watching water: From sky or stream? *Science*, *349*(6249), 684. https://doi.org/10.1126/science.349.6249.684-a

FAO. (2000, 2006). CROPWAT 8.0. Retrieved from http://www.fao.org/land-water/databases-and-software/cropwat/en/

Fernandez-Prieto, D., van Oevelen, P., Su, Z., & Wagner, W. (2012). Advances in Earth observation for water cycle science. *Hydrology and Earth System Sciences*, *16*, 543–549.

Foster, T., Brozović, N., Butles, A. P., Neale, C. M. U., Raes, D., Steduto, P., … Hsiao, T. C. (2017). AquaCrop-OS: An open source version of FAO's crop water productivity model. *Agricultural Water Management*, *181*, 18–22. https://doi.org/doi.org/10.1016/j.agwat.2016.11.015

Freeman, R. E. (1984). *Strategic Management: A stakeholder approach*. Boston: Pitman Publishing.

Gil Martínez, M. (2013). *Cebada y maíz rastrojero. Productividad económico-ambiental de la fertilización con purín (Barley and maize. Economic and environmental productivity of fertilisation with liquid manure) (*Informaciones técnicas No. 245) (p. 6). Zaragoza (Spain): Gobierno de Aragón. Departamento de Agricultura, Ganadería y Medio Ambiente. Dirección General de Alimentación y Fomento Agroalimentario. Servicio de Recursos Agrícolas. Retrieved from http://www.aragon.es/estaticos/GobiernoAragon/Departamentos/AgriculturaGanaderiaMedioAmbiente/AgriculturaGanaderia/Areas/07_Formacion_Inovacion_Sector_Agrario/02_Centro_Transferencia_Agroalimentaria/Publicaciones_Centro_Transferencia_Agroalimentaria/IT_2013/IT_245-13.pdf

Gutiérrez López, M. (2011). *Orientaciones varietales para las siembras de cereales en Aragón. Resultado de los ensayos. Cosecha 2011* (Informaciones Técnicas No. 230) (p. 28). Zaragoza (Spain): Diputación General de Aragón. Departamento de Agricultura y Alimentación. Dirección General de Desarrollo Rural, Servicio de Programas Rurales. Retrieved from http://aragon.es/estaticos/GobiernoAragon/Departamentos/AgriculturaGanaderiaMedioAmbiente/AgriculturaGanaderia/Areas/07_Formacion_Inovacion_Sector_Agrario/02_Centro_Transferencia_Agroalimentaria/Publicaciones_Centro_Transferencia_Agroalimentaria/IT_2011/IT_230-11.pdf

Gutiérrez López, M. (2016). *Transferencia de resultados de la red de ensayos de maíz y girasol en Aragón. Campaña 2015.* (Informaciones Técnicas No. 259) (p. 28). Zaragoza (Spain): Gobierno de Aragón. Departamento de Desarrollo Rural y Sostenibilidad. Dirección General de Desarrollo Rural. Servicio de Innovación y Transferenca Agroalimentaria. Retrieved from http://www.aragon.es/estaticos/GobiernoAragon/Departamentos/AgriculturaGanaderiaMedioAmbiente/TEMAS_AGRICULTURA_GANADERIA/Areas/FORMACION_INNOVACION_SECTOR_AGRARIO/CENTRO_TRANSFERENCIA_AGROALIMENTARIA/Publicaciones_Centro_Transferencia_Agroalimentaria/IT_2016/IT_259-16.pdf

Hall, D. K., Salomonson, V. V., & Riggs, G. A. (2006). *MODIS/Terra Snow Cover 8-Day L3 Global 500m Grid, Version 5 [h17-18v04]*. Boulder, Colorado USA: NASA National Snow and Ice Data Center Distributed Active Archive Center. Retrieved from http://dx.doi.org/10.5067/C574UGKQQU1T

Harrell, M. C., & Bradley, M. A. (2009). *Data collection methods. Semi-structured Interviews and focus groups* (p. 148). Santa Monica, CA (US): RAND.

Hernández-Mora, N., Gil, M., Garrido, A., & Rodríguez-Casado, R. (2013). *La sequía 2005-2008 en la cuenca del Ebro. Vulnerabilidad, Impactos y Medidas de Gestión* (p. 62). Universidad Politécnica de Madrid - Centro de Estudios e Investigación para la Gestión de Riesgos Agrarios y Medioambientales - CEIGRAM. Retrieved from http://www.ceigram.upm.es/sfs/otros/ceigram/LIBRO%20LA%20SEQUIA%20EN%20LA%20CUENCA%20DEL%20EBRO%20(1).pdf

Iglesias, A., & Garrote, L. (2015). Adaptation strategies for agricultural water management under climate change in Europe. *Agricultural Water Management*, *155*, 113–124. http://dx.doi.org/10.1016/j.agwat.2015.03.014

Iglesias, A., Sánchez, B., Garrote, L., & López, I. (2017). Towards Adaptation to Climate Change: Water for Rice in the Coastal Wetlands of Doñana, Southern Spain. *Water Resources Management*, *31*(2), 629–653. https://doi.org/10.1007/s11269-015-0995-x

Linés, C., Werner, M., & Bastiaanssen, W. (2017). The predictability of reported drought events and impacts in the Ebro Basin using six different remote sensing data sets. *Hydrol. Earth Syst. Sci.*, *21*, 4747–4765.

Lloveras, J., Martínez, E., & Santiveri, P. (2014). *Influencia de la fecha de siembra en el maíz en los regadíos del Valle del Ebro* (Vida Rural No. 36) (p. 5). Lleida (Spain): Agrotecnio. Universidad de Lleida.

Macauley, M. K. (2006). The value of information: Measuring the contribution of space-derived earth science data to resource management. *Space Policy*, *22*(4), 274–282. https://doi.org/10.1016/j.spacepol.2006.08.003

MAGRAMA. (2015). *Tecnichal-economical results of herbaceous crops 2014 (Resultados técnico-económicos de Cultivos Herbáceos 2014)*. Spain: Subdirección General de Análisis, Prospectiva y Coordinación, Subsecretaría. Ministerio de Agricultura, Alimentación y Medio Ambiente.

Matte, S., Boucher, M. A., Boucher, V., & Fortier Filion, T. C. (2017). Moving beyond the cost-loss ratio: Economic assessment of streamflow forecasts for a risk-Averse decision maker. *Hydrology and Earth System Sciences*, *21*(6), 2967–2986. https://doi.org/10.5194/hess-21-2967-2017

McMillan, H., Montanari, A., Cudennec, C., Savenije, H., Kreibich, H., Krueger, T., … Xia, J. (2016). Panta Rhei 2013–2015: global perspectives on hydrology, society and change. *Hydrological Sciences Journal*, 1–18. https://doi.org/10.1080/02626667.2016.1159308

Mylne, K. R. (2002). Decision-making from probability forecasts based on forecast value. *Meteorological Applications*, *9*(3), 307–315. https://doi.org/10.1017/S1350482702003043

O'Keeffe, J., Buytaert, W., Mijic, A., Brozović, N., & Sinha, R. (2016). The use of semi-structured interviews for the characterisation of farmer irrigation practices. *Hydrology and Earth System Sciences*, *20*(5), 1911–1924. https://doi.org/10.5194/hess-20-1911-2016

Perez y Perez, L., & Barreiro-Hurlé, J. (2009). Assessing the socio-economic impacts of drought in the Ebro River Basin. *Spanish Journal of Agricultural Research*, *7*(2), 269–280.

Quintilla, R., Portero, C., & Casterad, M. A. (2014). Apoyo a la gestión del agua en alta en la zona regable del Canal de Aragón y Cataluña con teledetección (p. 9). Presented atRemote sensing support to reservoir water management in Canal de Aragón y Cataluña irrigated area). In *Proceedings of* the XXXII Congreso Nacional de Riegos, (p. 9). Madrid. Retrieved from http://citarea.cita-aragon.es/citarea/bitstream/10532/2592/1/2014_156.pdf

Quiroga, S., Garotte, L., Iglesias, A., Fernández-Haddad, Z., Schlickenrieder, J., de Lama, B., … Sánchez-Arcilla, A. (2011). The economic value of drought information for water management under climate change: a case study in the Ebro basin. *Natural Hazards and Earth System Science*, *11*(1), 1–15. https://doi.org/10.5194/nhess-11-1-2011

Quiroga, S., Garrote, L., Iglesias, A., Fernández-Haddad, Z., Schlickenrieder, J., de Lama, B., … Sánchez-Arcilla, A. (2011). The economic value of drought information for water management under climate change: a case study in the Ebro basin. *Natural Hazards and Earth System Sciences*, *11*, 1–15.

R Core Team. (2016). *R: A language and environment for statistical computing*. Viena, Austria: R Foundation for Statistical Computing. Retrieved from https://www.R-project.org/.

Roulin, E. (2006). Skill and relative economic value of medium-range hydrological ensemble predictions. *Hydrology and Earth System Sciences Discussions*, *3*(4), 1369–1406. https://doi.org/10.5194/hessd-3-1369-2006

Sivapalan, M., Savenije, H. H. G., & Blöschl, G. (2012). Socio-hydrology: A new science of people and water. *Hydrological Processes*, *26*(8), 1270–1276. https://doi.org/10.1002/hyp.8426

Stanski, H. R., Wilson, L. J., & Burrows, W. R. (1989). *Survey of common verification methods in meteorology* (Research report No. MSRB 89-5) (p. 81). Downsview, Ontario (Canada): Atmospheric Environment Service. Retrieved from http://www.cawcr.gov.au/projects/verification/Stanski_et_al/Stanski_et_al.html

van Dijk, A. I. J. ., & Renzullo, L. J. (2011). Water resource monitoring system and the role of satellite observations. *Hydrology and Earth System Sciences*, *15*, 39–55. https://doi.org/10.5194/hess-15-39-2011

Van Loon, A. F., Gleeson, T., Clark, J., Van Dijk, A. I. J. M., Stahl, K., Hannaford, J., … van Lanen, H. A. J. (2016). Drought in the Anthropocene. *Nature Geoscience*, *9*, 89–91.

Verkade, J. S., & Werner, M. G. F. (2011). Estimating the benefits of single value and probability forecasting for flood warning. *Hydrology and Earth System Sciences*, *15*,(12), 3751–3765. https://doi.org/10.5194/hess-15-3751-2011

Williamson, R. A., Hertzfeld, H. R., Cordes, J., & Logsdon, J. M. (2002). The socioeconomic benefits of Earth science and applications research: reducing the risks and costs of natural disasters in the USA. *Space Policy*, *18*(1), 57–65. https://doi.org/10.1016/S0265-9646(01)00057-1

Zhu, Y., Toth, Z., Wobus, R., Richardson, D., & Mylne, K. (2002). The economic value of ensemble-based weather forecasts. *Bulletin of the American Meteorological Society, 83*(1), 73–83. https://doi.org/10.1175/1520-0477(2002)083<0073:TEVOEB>2.3.CO;2

---

## Author Response (AR2)

We would like to thank the editor for taking the time to review the manuscript and suggesting improvements. Please, find below our replies to the comments and the changes made in the manuscript to address them.

**Editor comment 1: Fig 4. In the legend, please explain the meaning of the colours and the grey boxes. Please indicate where the time 'i' lies on the timeline. Please put the 'period' legend in chronological order.In the legend, please explain the meaning of the colours and the grey boxes.**

The following explanation of the coloured parameters and the grey boxes has been added to the caption of Figure 4: "The parameters representing the information that is exchanged between the two decision models are coloured in green (Crop Surface, CS) and blue (Curtailments, Cu). The grey boxes represent different blocks of the decision models. The reservoir operation decision has the same kind of input and output for each decision date, while the farmer's decision has different inputs before and after the start of the irrigation season. The white lines within the blocks represent the moments at which decisions are made (Time i)".

The 'Period' legend has been reordered chronologically and it has been renamed as 'Time', since the elements in that legend represent points in time rather than periods of time.

**Editor comment 2:  Table 1/Fig3. Please identify the paths 1-7 on Fig 3.**

The paths in Table 1 are a combination of the decisions of the different types of farmers. To reference this in Table 1, the courses of action of each farmer type have been labelled with lower case letters on the right side of Figure 3. Table 1 has been expanded to include a reference to those courses of action and the crops planted as a result of following them.

The captions of both Figure 3 and Table 1 have been updated to describe the new labels and columns.

**Editor comment 3:  Fig 7. Does this show value relative to the uninformed or perfect scenario?**

The Relative Value in Figure 7 is calculated with the equation described in section 5, which includes both the uninformed and perfect information scenarios:

RV = (Value_information – Value_uninformed) / (Value_perfect – Value_uninformed)

We added a reference to the equation in the text that describes Figure 7 to make it clearer.

**Editor comment 4: Fig 8. Please define the axes abbreviations in the legend so this figure can be read without reference to the previous figure(s).**

The caption of Figure 8 has been edited to include the axes abbreviations.

**Modified figures:**

[Figure]

| Parameters | Time [ ] | Availability and sources |
|---|---|---|
| F – Inflow | h – start of hydrological season | **V** known to decision maker (from data) |
| D – Demand | i – decision date | ☐ known to decision maker (from model) |
| V – Volume | s – start of irrigation season | ◯ unknown to decision maker (estimated) |
| CS – Crop surface | n – end of season | |
| CD – Crop demand | | |
| Cu - Curtailments | [s:i] from s to i | |

*Figure 1. Model decisions, parameters that inform them and decision outputs. A description of the abbreviations is included below. The period considered for each parameter is given in between square brackets. The shapes indicate the availability and source of the information. The parameters representing the information that is exchanged between the two decision models are coloured in green (Crop Surface, CS) and blue (Curtailments, Cu). The grey boxes represent different blocks of the decision models. The reservoir operation decision has the same kind of input and output for each decision date, while the farmer's decision has different inputs before and after the start of the irrigation season. The white lines within the blocks represent the moments at which decisions are made (Time i).*

[Figure]

*Figure 2. Crop options considered in the model for farmers. Blue and red As represent respectively good and poor water availability at the moment of the decision. R1, R2 and R3 mark the different courses of action that farmers can follow depending on the risk they are willing to take, with R1 being the most risk averse and R3 the least risk averse. The lower case letters a-h indicate the end points of the possible decision paths. The blue vertical line marks the start of the irrigation season.*

| Path | Decision moments | | | | Decision by farmer type | | | |
|------|-----|-----|-----|-----|-----|-------|-------|-------|
| | Nov | Feb | Apr | May | T1 | T2-R1 | T2-R2 | T2-R3 |
| 1 | 🟥 | 🟥 | 🟥 | 🟥 | d | e | f | h |
| 2 | 🟥 | 🟥 | 🟦 | 🟥 | d | e | f | g |
| 3 | 🟥 | 🟦 | 🟨 | 🟥 | d | e | g | g |
| 4 | 🟥 | 🟥 | 🟥 | 🟦 | c | e | f | h |
| 5 | 🟥 | 🟥 | 🟦 | 🟦 | c | e | f | g |
| 6 | 🟥 | 🟦 | 🟨 | 🟦 | c | e | g | g |

[revised manuscript text omitted]
* (pp. 571–580). Zaragoza: Universidad de Zaragoza-AGE. Retrieved from http://congresoage.unizar.es/eBook/trabajos/060_Casterad%20Seral.pdf

CHE. (n.d.). Portal de CHEbro. Retrieved 15 March 2018, from http://www.chebro.es/contenido.visualizar.do?idContenido=37945&idMenu=2167

Eden, C., & Ackermann, F. (1998). *Making strategy: the journey of strategy management*. London: SAGE Publications Ltd.

Espluga Trenc, J. (2016). What is happening with transgenic maize? (¿Qué está pasando con el cultivo de maíz transgénico?). *Soberanía alimentaria*, *26*, 34–38.

Famiglietti, J. S., Cazenave, A., Eicker, A., Reager, J. T., Rodell, M., Velicogna, I., … Zhulidov, A. V. (2015). Watching water: From sky or stream? *Science*, *349*(6249), 684. https://doi.org/10.1126/science.349.6249.684-a

FAO. (2000, 2006). CROPWAT 8.0. Retrieved from http://www.fao.org/land-water/databases-and-software/cropwat/en/

Fernandez-Prieto, D., van Oevelen, P., Su, Z., & Wagner, W. (2012). Advances in Earth observation for water cycle science. *Hydrology and Earth System Sciences*, *16*, 543–549.

Foster, T., Brozović, N., Butles, A. P., Neale, C. M. U., Raes, D., Steduto, P., … Hsiao, T. C. (2017). AquaCrop-OS: An open source version of FAO's crop water productivity model. *Agricultural Water Management*, *181*, 18–22. https://doi.org/doi.org/10.1016/j.agwat.2016.11.015

Freeman, R. E. (1984). *Strategic Management: A stakeholder approach*. Boston: Pitman Publishing.

Gil Martínez, M. (2013). *Cebada y maíz rastrojero. Productividad económico-ambiental de la fertilización con purín (Barley and maize. Economic and environmental productivity of fertilisation with liquid manure)* (Informaciones técnicas No. 245) (p. 6). Zaragoza (Spain): Gobierno de Aragón. Departamento de Agricultura, Ganadería y Medio Ambiente. Dirección General de Alimentación y Fomento Agroalimentario. Servicio de Recursos Agrícolas. Retrieved from http://www.aragon.es/estaticos/GobiernoAragon/Departamentos/AgriculturaGanaderiaMedioAmbiente/AgriculturaGanaderia/Areas/07_Formacion_Inovacion_Sector_Agrario/02_Centro_Transferencia_Agroalimentaria/Publicaciones_Centro_Transferencia_Agroalimentaria/IT_2013/IT_245-13.pdf

Gutiérrez López, M. (2011). *Orientaciones varietales para las siembras de cereales en Aragón. Resultado de los ensayos. Cosecha 2011* (Informaciones Técnicas No. 230) (p. 28). Zaragoza (Spain): Diputación General de Aragón. Departamento de Agricultura y Alimentación. Dirección General de Desarrollo Rural, Servicio de Programas Rurales. Retrieved from http://aragon.es/estaticos/GobiernoAragon/Departamentos/AgriculturaGanaderiaMedioAmbiente/AgriculturaGanaderia/Areas/07_Formacion_Inovacion_Sector_Agrario/02_Centro_Transferencia_Agroalimentaria/Publicaciones_Centro_Transferencia_Agroalimentaria/IT_2011/IT_230-11.pdf

Gutiérrez López, M. (2016). *Transferencia de resultados de la red de ensayos de maíz y girasol en Aragón. Campaña 2015.* (Informaciones Técnicas No. 259) (p. 28). Zaragoza (Spain): Gobierno de Aragón. Departamento de Desarrollo Rural y Sostenibilidad. Dirección General de Desarrollo Rural. Servicio de Innovación y Transferenca Agroalimentaria. Retrieved from http://www.aragon.es/estaticos/GobiernoAragon/Departamentos/AgriculturaGanaderiaMe

dioAmbiente/TEMAS_AGRICULTURA_GANADERIA/Areas/FORMACION_INNOVACION_SECTO
R_AGRARIO/CENTRO_TRANSFERENCIA_AGROALIMENTARIA/Publicaciones_Centro_Transfer
encia_Agroalimentaria/IT_2016/IT_259-16.pdf

Hall, D. K., Salomonson, V. V., & Riggs, G. A. (2006). *MODIS/Terra Snow Cover 8-Day L3 Global 500m Grid, Version 5 [h17-18v04]*. Boulder, Colorado USA: NASA National Snow and Ice Data Center Distributed Active Archive Center. Retrieved from http://dx.doi.org/10.5067/C574UGKQQU1T

Harrell, M. C., & Bradley, M. A. (2009). *Data collection methods. Semi-structured Interviews and focus groups* (p. 148). Santa Monica, CA (US): RAND.

Hernández-Mora, N., Gil, M., Garrido, A., & Rodríguez-Casado, R. (2013). *La sequía 2005-2008 en la cuenca del Ebro. Vulnerabilidad, Impactos y Medidas de Gestión* (p. 62). Universidad Politécnica de Madrid - Centro de Estudios e Investigación para la Gestión de Riesgos Agrarios y Medioambientales - CEIGRAM. Retrieved from http://www.ceigram.upm.es/sfs/otros/ceigram/LIBRO%20LA%20SEQUIA%20EN%20LA%20C UENCA%20DEL%20EBRO%20(1).pdf

Iglesias, A., & Garrote, L. (2015). Adaptation strategies for agricultural water management under climate change in Europe. *Agricultural Water Management*, *155*, 113–124. http://dx.doi.org/10.1016/j.agwat.2015.03.014

Iglesias, A., Sánchez, B., Garrote, L., & López, I. (2017). Towards Adaptation to Climate Change: Water for Rice in the Coastal Wetlands of Doñana, Southern Spain. *Water Resources Management*, *31*(2), 629–653. https://doi.org/10.1007/s11269-015-0995-x

Linés, C., Werner, M., & Bastiaanssen, W. (2017). The predictability of reported drought events and impacts in the Ebro Basin using six different remote sensing data sets. *Hydrol. Earth Syst. Sci.*, *21*, 4747–4765.

Lloveras, J., Martínez, E., & Santiveri, P. (2014). *Influencia de la fecha de siembra en el maíz en los regadíos del Valle del Ebro* (Vida Rural No. 36) (p. 5). Lleida (Spain): Agrotecnio. Universidad de Lleida.

Macauley, M. K. (2006). The value of information: Measuring the contribution of space-derived earth science data to resource management. *Space Policy*, *22*(4), 274–282. https://doi.org/10.1016/j.spacepol.2006.08.003

MAGRAMA. (2015). *Tecnichal-economical results of herbaceous crops 2014 (Resultados técnico-económicos de Cultivos Herbáceos 2014)*. Spain: Subdirección General de Análisis, Prospectiva y Coordinación, Subsecretaría. Ministerio de Agricultura, Alimentación y Medio Ambiente.

Matte, S., Boucher, M. A., Boucher, V., & Fortier Filion, T. C. (2017). Moving beyond the cost-loss ratio: Economic assessment of streamflow forecasts for a risk-Averse decision maker. *Hydrology and Earth System Sciences*, *21*(6), 2967–2986. https://doi.org/10.5194/hess-21-2967-2017

McMillan, H., Montanari, A., Cudennec, C., Savenije, H., Kreibich, H., Krueger, T., … Xia, J. (2016). Panta Rhei 2013–2015: global perspectives on hydrology, society and change. *Hydrological Sciences Journal*, 1–18. https://doi.org/10.1080/02626667.2016.1159308

Mylne, K. R. (2002). Decision-making from probability forecasts based on forecast value. *Meteorological Applications*, *9*(3), 307–315. https://doi.org/10.1017/S1350482702003043

O'Keeffe, J., Buytaert, W., Mijic, A., Brozović, N., & Sinha, R. (2016). The use of semi-structured interviews for the characterisation of farmer irrigation practices. *Hydrology and Earth System Sciences*, *20*(5), 1911–1924. https://doi.org/10.5194/hess-20-1911-2016

Perez y Perez, L., & Barreiro-Hurlé, J. (2009). Assessing the socio-economic impacts of drought in the Ebro River Basin. *Spanish Journal of Agricultural Research*, *7*(2), 269–280.

Quintilla, R., Portero, C., & Casterad, M. A. (2014). Apoyo a la gestión del agua en alta en la zona regable del Canal de Aragón y Cataluña con teledetección (Remote sensing support to reservoir water management in Canal de Aragón y Cataluña irrigated area). In *Proceedings of*

the *XXXII Congreso Nacional de Riegos* (p. 9). Madrid. Retrieved from http://citarea.cita-aragon.es/citarea/bitstream/10532/2592/1/2014_156.pdf

Quiroga, S., Garotte, L., Iglesias, A., Fernández-Haddad, Z., Schlickenrieder, J., de Lama, B., … Sánchez-Arcilla, A. (2011). The economic value of drought information for water management under climate change: a case study in the Ebro basin. *Natural Hazards and Earth System Science*, *11*(1), 1–15. https://doi.org/10.5194/nhess-11-1-2011

Quiroga, S., Garrote, L., Iglesias, A., Fernández-Haddad, Z., Schlickenrieder, J., de Lama, B., … Sánchez-Arcilla, A. (2011). The economic value of drought information for water management under climate change: a case study in the Ebro basin. *Natural Hazards and Earth System Sciences*, *11*, 1–15.

R Core Team. (2016). *R: A language and environment for statistical computing*. Viena, Austria: R Foundation for Statistical Computing. Retrieved from https://www.R-project.org/.

Roulin, E. (2006). Skill and relative economic value of medium-range hydrological ensemble predictions. *Hydrology and Earth System Sciences Discussions*, *3*(4), 1369–1406. https://doi.org/10.5194/hessd-3-1369-2006

Sivapalan, M., Savenije, H. H. G., & Blöschl, G. (2012). Socio-hydrology: A new science of people and water. *Hydrological Processes*, *26*(8), 1270–1276. https://doi.org/10.1002/hyp.8426

Stanski, H. R., Wilson, L. J., & Burrows, W. R. (1989). *Survey of common verification methods in meteorology* (Research report No. MSRB 89-5) (p. 81). Downsview, Ontario (Canada): Atmospheric Environment Service. Retrieved from http://www.cawcr.gov.au/projects/verification/Stanski_et_al/Stanski_et_al.html

Van Loon, A. F., Gleeson, T., Clark, J., Van Dijk, A. I. J. M., Stahl, K., Hannaford, J., … van Lanen, H. A. J. (2016). Drought in the Anthropocene. *Nature Geoscience*, *9*, 89–91.

Verkade, J. S., & Werner, M. G. F. (2011). Estimating the benefits of single value and probability forecasting for flood warning. *Hydrology and Earth System Sciences*, *15*(12), 3751–3765. https://doi.org/10.5194/hess-15-3751-2011

Williamson, R. A., Hertzfeld, H. R., Cordes, J., & Logsdon, J. M. (2002). The socioeconomic benefits of Earth science and applications research: reducing the risks and costs of natural disasters in the USA. *Space Policy*, *18*(1), 57–65. https://doi.org/10.1016/S0265-9646(01)00057-1

Zhu, Y., Toth, Z., Wobus, R., Richardson, D., & Mylne, K. (2002). The economic value of ensemble-based weather forecasts. *Bulletin of the American Meteorological Society*, *83*(1), 73–83. https://doi.org/10.1175/1520-0477(2002)083<0073:TEVOEB>2.3.CO;2